# Natural Hydroxybenzoic and Hydroxycinnamic Acids Derivatives: Mechanisms of Action and Therapeutic Applications

**DOI:** 10.3390/antiox14060711

**Published:** 2025-06-11

**Authors:** Sergio López-Herrador, Julia Corral-Sarasa, Pilar González-García, Yaco Morillas-Morota, Enrica Olivieri, Laura Jiménez-Sánchez, María Elena Díaz-Casado

**Affiliations:** 1Departamento de Fisiología, Facultad de Medicina, Universidad de Granada, 18016 Granada, Spain; sergiolopez@ugr.es (S.L.-H.); juliacorral@ugr.es (J.C.-S.); pgonzalez@ugr.es (P.G.-G.); yacomorillas@correo.ugr.es (Y.M.-M.); olivierie@ugr.es (E.O.); ext.ljimsan@ugr.es (L.J.-S.); 2Instituto de Investigación Biosanitaria ibs.Granada, 18016 Granada, Spain

**Keywords:** hydroxybenzoic acid, hydroxycinnamic acid, therapeutic applications

## Abstract

Hydroxybenzoic and hydroxycinnamic acid derivatives are a class of organic compounds derived from benzoic and cinnamic acid, respectively, two types of acids in which one of the hydrogen atoms on the benzene ring is replaced by a hydroxyl (-OH) or alkoxyl group (-O-R). These compounds are found in a wide range of natural sources, particularly in plants, as well as in some fungi. Their biological properties are distinctive, as they combine antioxidant, anti-inflammatory, and metabolic functions, which can be utilized for therapeutic purposes in humans. In this review, we identify the most relevant hydroxybenzoic and hydroxycinnamic acid derivatives for human applications, explain their common and specific mechanisms of action, and highlight their applications in significant preclinical and clinical studies.

## 1. Introduction

Phenolic acids have been gathering interest as potential therapeutic agents in a wide variety of pathologies, as they exhibit antioxidant, anti-inflammatory, antitumoral, and antimicrobial properties [1,2,3,4,5,6], among others. Likewise, epidemiological studies provide supporting evidence linking a higher dietary intake of these compounds to numerous health benefits [7], including a reduced risk of suffering from cardiovascular disease [5,8,9], prevention of certain types of cancer [10], and metabolic disorders [4].

From a structural perspective, phenolic acids and their derivatives fall under two main groups: hydroxybenzoic acids and hydroxycinnamic acids (Table 1) [5,10]. These groups are structurally derived from the non-phenolic molecules of benzoic and cinnamic acid, respectively. They both share a hydroxyl (-OH) substituted benzene ring (phenol), which stabilizes the molecule by resonance [4], along with a carboxyl (-COOH) functional group. However, they differ in the position of the carboxyl group; in hydroxybenzoic acids, it is directly bonded to the phenol ring, whereas in hydroxycinnamic acids, it is separated by a carbon–carbon double bond [1,3].

Derivatives of these compounds are formed through further modifications in the benzene ring, usually by the addition of hydroxyl or alkoxyl (-O-R) substituents [1,5,8,11].

Some of the most studied hydroxybenzoic acids include 4-hydroxybenzoic acid (4-HB), *β*-resorcylic acid (BRA), vanillic acid (VA), and gallic acid [3]. While most common hydroxycinnamic acids are caffeic acid (CA), ferulic acid (FA), sinapic acid (SA), and *p*-coumaric acid (*p*-CA) [11].

The diverse structures within the phenolic acid family are crucial in determining their activity within the organism [4,5]. Specifically, the number and position of hydroxyl groups have been identified as determinant factors of their antioxidant capacity [5,7,8], enabling them to directly interact with free radicals via single-electron transfer or hydrogen-atom transfer [4] or indirectly inducing the expression of endogenous enzymes with an antioxidant effect [2] (Figure 1). Additionally, phenolic acids can inhibit the production of pro-inflammatory cytokines and modulate signaling pathways present during the inflammatory response (i.e., NF- κB pathway) [5,6,7,9]. They can also interfere with pathogen activity by acidifying the cytoplasm [2,4], inhibit enzymes such as cyclooxygenases [8,9,10], interact with different receptors modulating their expression, like increasing the expression of the GLUT-2 insulin receptors [2], and even act as bypass molecules in certain metabolic deficiencies (i.e., primary coenzyme Q deficiency) (Figure 2) [12]. Consequently, different phenolic acids exhibit a range of biological activities (Table 1).

**Table 1 antioxidants-14-00711-t001:** Effects of hydroxybenzoic and hydroxycinnamic acids. Overview of the reported therapeutic effects of the mentioned hydroxybenzoic and hydroxycinnamic acids and the mechanism of action through which they achieve them. Their chemical structure and alternative nomenclature.

Molecule	Therapeutic Effect	Mechanism of Action
**Hydroxybenzoic Acids**		
**4-Hydroxybenzoic acid** 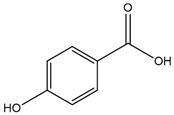	Anti-inflammatory	Inhibition of the Nlrp3 inflammasome via ROS elimination [13]
		Inhibition of IL-1β release [13].
		Reduction in Asc protein aggregation [13]
		Reduction in Il-1β, Tnf, Il-6, Nlrp3, and Casp1 transcription [13]
		Regulation of Myd88 signalling pathway [13]
		Reduction pro-inflammatory cytokines levels (IL-4, IL-6, TNF-α) [14,15]
		Increase in anti-inflammatory cytokine levels (IL-10) [14,15]
		Reduction in LPS-induced systemic inflammation by lowering IL-1β levels [13]
		Reduction in the proportion of Th17 and Treg inflammatory cells [14]
	Antioxidant	Inhibition of free radicals [16]
		Free radical scavenging [16]
	Antihypertensive	Linked to antioxidant activity and interaction with autonomic ganglia and muscarinic receptors [16]
	Antitumour	Modulation of the PI3K/Akt, MAPK3, STAT3 metabolic pathways [17]
	Primary CoQ deficiency improvement	Serves as a substrate for the COQ2 enzyme and can restore CoQ levels in COQ2 pathogenic variants [18,19]
	Intestinal barrier and microbiome modulation	Reversal of MUC2 reduction (goblet cells increase) [14]
		Restoration of *Muc1*, *Muc2*, *Muc3* expression related to mucin production [14]
		Increase in *Akkermansia muciniphila* abundance [14]
	Metabolic regulation	Insulin secretion enhancement [20]
		Modulation of GLUT4 expression [20]
		Activation of PPARγ [20]
		Dual agonism of PPARγ and GPR40 receptors (in silico) [20]
	Neuroprotection	Reduction in toxic αS aggregate form [21,22]
		Inhibition of intracellular and cell-to-cell αS transmission [21,22]
	Antimicrobial	Inhibition of zoospore motility and cystospore germination of *Phytophthora sojae* [23]
	Antifungal-antibiotic production	Enhancement of HSAF biosynthesis in *Lysobacter enzymogenes* via LysRLe regulation of LenB2 enzyme [24]
***ß*-Resorcylic acid** (2,4-Dihydroxybenzoic acid) * 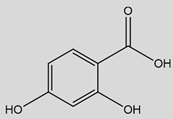 *	Anti-neuroinflammatory	Decrease in CXCL10 and CCL2 chemokines [25,26]
		Reduction in GFAP expression and reactive microglial cells [26,27]
		Promotion of a microglial phenotype shift from a pro-inflammatory to a relaxed state [26,27]
		Modulation of inflammation related genes expression (*Bgn, Ccl6, Cst7, Ifi27l2a, Ifitm3, Vav1)* [27]
		Normalisation of immune related proteins plasma levels (SERPINA, MASP1, AI182371) and brain metabolites N-AC-Glu and N-Ac-Glu-6P [26]
		Anti-inflammatory effects are unrelated to direct NF-κB action [27]
	Antioxidant	Free radical scavenger and single electron transfer capture [28]
	Antimicrobial	Potent activity against Gram-negative (*E. coli*, *P. multocida,* or *N. gonorrhoeae*) and Gram-positive (*S. aureus* and *E. faecalis*) [25,29]
	CoQ deficiency improvement	Bypass effect in *Coq7* and *Coq9* deficiencies as BRA contains the hydroxyl group added by COQ7/COQ9 [26,27,30,31,32]
		Improvement of mitochondrial CoQ levels, reduction in toxic DMQ accumulation, and stabilization of Q complex [27,31,32]
		Improvement of mitochondrial bioenergetics (esp. liver, brain, kidneys) [27,31]
		Normalisation of the mitochondrial proteome (CoQ-dependent enzymes, β-oxidation, folate/glycine metabolism, nucleotide biosynthesis, TCA cycle, carnitine shuttle, and OxPhos system) [26]
		Potential CoQ-independent mechanisms in *Adck4* and *Coq6* mutation models with reversal of the pathogenic phenotype post administration (mechanisms poorly understood) [33,34]
		Secondary CoQ deficiency restoration by CoQ mitochondrial metabolism modulation [35]
	Metabolic regulation	Restoration of mitochondrial CoQ metabolism in WAT, reducing adipocyte hypertrophy [36]
		Metabolic remodelling via HFN4α/LXR-dependent towards enhanced lipid catabolism [35]
		Prevention of ectopic fat accumulation [35]
		Synergistic effects (WAT CoQ normalization + hepatic lipid catabolism) enhance glucose homeostasis [35]
	Antitumour agent	Inhibition of CDK1, arresting cell cycle progression [37,38]
**Vanillic acid** (4-Hydroxy-3-methoxybenzoic acid) 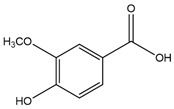	Anti-inflammatory	Suppression of pro-inflammatory cytokines production (TNF-α, IL-6, IL-1β) via NF-κB inhibition [39,40,41,42]
		Modulation of cytokines (↓CXCL10, ↓CCL2 and ↑IL-6, ↑IL-10 trends [26]
		Downregulation of COX-2 and iNOS [40,41,42,43]
		Modulation of MAPK and JAK/STAT pathways [40,41,42,43]
		Reduction of glial cell activation [26]
		Inhibition of ferroptosis [44]
		Inhibition of inflammatory mediator production [45]
		Reduction of caspase-1 activity in mast cells and suppression of MAPK phosphorylation [46]
		Reduction in NLRP3 inflammasome in synovial tissue [23]
		Inhibition of neutrophil recruitment [47]
	Antioxidant	Direct free radical scavenging (ROS, RNS, H_2_O_2_, HOCl) [39,48,49,50]
		Activation of the AMPK signalling pathway [51]
		Enhancement of endogenous antioxidant enzymes (GSH, SOD, GPx, CAT) [52,53,54,55]
		Inhibition of lipid peroxidation (TBARS and protein-bound carbonyls formation prevention) [39,56]
		Regulation of mitophagy (↑PINK1/Parkin/Mfn2 proteins, ↑LC3-II/LC3-I ratio, and ↓p62 levels) [57]
	CoQ deficiency improvement	Bypass the effect of CoQ biosynthesis over the COQ6 enzyme [12]
		Improvement through a non-bypass mechanism over the C*oq2* model [58] and *COQ9* fibroblasts [59]
		Decrease in DMQ/CoQ ratio in peripheral tissues, increase in mitochondrial bioenergetics [26]
		Normalisation of the mitochondrial proteome and metabolism [26]
		COQ4 overexpression [18,26]
	Metabolic regulation	Improvement of insulin sensitivity; reduction in fasting glucose, insulin, and blood pressure [60]
		Enhancement of antioxidant status (↑SOD, CAT, GPx, GSH, vitamins C and E activities, and ↓lipid peroxidation) [60]
		Inhibition of the PTP1B enzyme [61]
		Activation of AMPKSirt1/PGC-1α pathway [62]
		Modulation of insulin signalling pathway (Akt, ERK1/2) [62]
		Regulation of glucose and lipid metabolism enzymes [62]
		Improvement of lipid profile (↑HDL-C, ↓Chol/TG/FFA/LDL-C/VLDL-C) [62]
		Lipid modulation (↓HMG-CoA and ↑LCAT activities) [62]
		Adipogenesis suppression (↓PPARγ, C/EBPα; ↑AMPKα regulation) [63]
		Inhibition of lipid accumulation [63]
		Enhancement of thermogenesis in BAT [63]
		Anti-obesity effect is controversial [35]
		Hepatoprotective (mitigates mitochondrial dysfunction via ↑AMPK/Sirt1/PGC-1α) [64,65,66]
	Antitumour	Induction of mitochondrial apoptosis (G1 phase arrest, inhibiting proliferation) [67]
		Enhancement of chemotherapy efficacy (mechanism poorly understood) [67]
		Reduction in TBARS, lipid hydroperoxides, and CYP450; increase in antioxidant levels in plasma and uterus [68]
		Downregulation of MMP-2, MMP-9, and cyclin D1 expression [68]
		Increase in apoptosis/autophagy markers [69]
		Repression of STAT3 phosphorylation [69]
	Neuroprotection	Attenuation of cerebral reactive hyperaemia [70]
		Protection against blood-brain barrier disruption [70]
		Reduction in anxiety-like behaviours [70]
		Myelination promotion [71]
	Bone Health Promotion	Stimulation of osteoblast proliferation and enhancement of bone formation marker expression (via MAP kinase/ER signalling) [72]
	Antimicrobial	Inhibition of growth, biofilm, virulence in Gram-positive and Gram-negative bacteria; enhances synthetic antibiotic effects against ESKAPE pathogens [73]
**Protocatechuic Acid** (3,4-Dihydroxybenzoic acid) 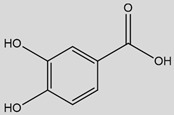	Anti-Inflammatory	Inhibition of NF-κB signalling (blockage of IκB-α degradation/p65 phosphorylation) [74,75]
		Reduction in proinflammatory gene expression (TNF-α and IL-1β) [75,76,77,78]
		Interference with the MAPK pathway (p38, c-Jun N-terminal kinase/JNK, and ERK1/2 phosphorylation inhibition) [75]
		Targeting of TLR4 signalling (downregulation [77], suppression of Akt, mTOR, JNK, p38 [79])
		Downregulation of pro-inflammatory mediators; ↓TNF-α, IL-1β, IL-6 [75,77]
		Reduction in PGE_2_ and NO via ↓COX-2 and iNOS [75,80,81]
		Inhibition of leukocyte recruitment (↓VCAM-1, ICAM-1 expression/secretion) and monocyte migration impairment (↓CCR2 expression; ↓monocyte adhesion/infiltration) [82,83,84,85]
		Activation of SIRT1 pathway (inhibits NF-κB via deacetylation, IKKβ)—(↓pro-inflammatory markers and ↑PPARγ) [86,87]
		Induction of HO-1 via Nrf2 activation [88,89,90]
	Neuroprotection (via anti-neuroinflammatory effect)	Modulation of glial activation via M1 inhibition and M2 shift, along with cytokine reduction [91,92]
		Attenuation of microglial and astrocyte activation in the hippocampus, preserving blood–brain barrier integrity [93]
		Reduction in oxidative damage [93]
	Antioxidant	Neutralisation of ROS via catechol group H/electron donation [94]
		Potent scavenging activity (aqueous and lipid environment) [94]
		Chelation of transition metal ions (like Fe^2+^ and Cu^2+^) [95]
		Activation of Nrf2 pathway and subsequent upregulation of antioxidant enzymes (HO-1, SOD, CAT, GPx) [89,90,96,97,98]
		Maintenance of GSH reduced levels [97,99]
		Inhibition of lipid peroxidation (peroxyl radicals scavenging, membrane stabilisation, MDA/TBARS markers reduction) [86,87,96,100,101,102]
	Hepatic protection	Reduction in inflammatory cell infiltration, congestion, and liver swelling [103]
		Decrease in hepatic MDA [76]
		Mitigation of endoplasmic reticulum stress [104]
		Modulation of oxidative stress markers (↓TBARS and lipid profile improvement) [105]
	Cardiovascular protection	Reduction of VCAM-1 secretion in endothelial cells [84]
		Suppression of monocyte adhesion (↓NF-κB activity) restrains atherosclerotic development [82]
	CoQ deficiency improvement	Bypass the effect of CoQ biosynthesis over the COQ6 enzyme [106,107]
**Hydroxycinnamic acids**		
***p*-Coumaric acid** (4-Hydroxycinnamic acid) 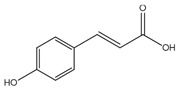	Antioxidant	Neutralisation of free radicals (Enhancement of fatty acid oxidation) [108,109,110]
	Membrane potential modulation	Modulation of electrical potential affecting cellular signalling [111]
	Anti-inflammatory	Reduction in pro-inflammatory cytokine production [112,113].
	Metabolic regulation	Reduction in adipokine production (insulin resistance association) [112,113].
	Antitumour	Induction of apoptosis and angiogenesis suppression [114,115,116,117,118]
**Caffeic acid** (3,4-Dihydroxycinnamic acid) 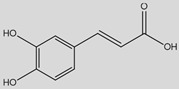	Antioxidant	Enhancement of antioxidant enzymes (GPx, SOD) and ROS production decrease [119,120,121,122,123,124,125,126,127,128,129,130,131,132]
	Antitumour	Metastasis inhibition by EMT suppression and modulation of PI3K/Akt and AMPK signalling pathways [119,133]
	Anti-inflammatory	Inhibition of pro-inflammatory cytokine release [134,135]
	Neuroprotection	Regulation of microglial activation in the hippocampus [134,135]
**Ferulic acid** (4-Hydroxy-3-methoxycinnamic acid) 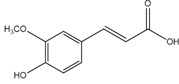	Antioxidant	Scavenging of free radicals and upregulation of cytoprotective systems [136,137,138,139,140]
	Antitumour	Protection against UV damage and carcinogenesis [136,137,138,139,140]
	Anti inflammatory	Inhibition of proinflammatory cytokines production and regulation of NF-κB and p38 MAPK signalling [141,142,143,144,145]
	Cardiovascular risk	Reduction of platelet aggregation [146,147,148,149,150]
**Sinapic acid** (4-Hydroxy-3,5-dimethoxycinnamic acid) 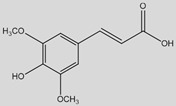	Antioxidant	Protection from lysosome dysfunction and oxidative damage by free radical scavenging and antioxidant enzyme activity enhancement [151,152,153]
	Anti-inflammatory	Suppression of T-helper 2 immune response [154,155,156]
	Metabolic regulation	Modulation of lipid metabolism [116,146,156]
	Antitumour	Promotion of apoptosis in cancer cells by increasing caspase-3 activity, and cell invasion inhibition [157,158]

Regarding their distribution, these compounds are predominantly found in plants and some fungi [3], often in bound forms like amides, esters, or glycosides, and less frequently in their free form [2,7,11]. The biosynthesis of these secondary metabolites primarily follows the shikimate pathway, originating from L-phenylalanine or L-tyrosine [3], and can also be produced, although less important, from the mevalonic acid pathway in higher plants [5]. Their production will be triggered as a response to biotic or abiotic stress [4], providing a protective function against ultraviolet light, insects, viruses, and bacteria [3] that would explain their inherent bioactivity.

Broadly speaking, phenolic acids tend to be found as part of our diets in fruits, vegetables, mushrooms, and some beverages. As an example, berries are rich in caffeic acid and 4-HB, and citrus fruits are rich in cinnamic acid derivatives like *p*-coumaric acid and ferulic acid [7,11]. Mangoes are sources of gallic, protocatechuic, chlorogenic, and VA [7]. In vegetables, these acids are concentrated in seeds, skins, and leaves [2]. Spices like star anise, rich in protocatechuic acid, also contribute to our intake of these compounds [9]. Whole grains also have a high content of phenolic acids, examples include wheat and rye, which show gentisic acid [9].

Despite the established broad therapeutic potential of phenolic acids, a focused and comprehensive review specifically addressing the mechanisms of action and therapeutic applications of hydroxybenzoic acid derivatives is warranted. These compounds, with their unique structural features, exhibit distinct biological activities that hold significant promise for addressing various human health conditions. Therefore, this review aims to identify and describe the key hydroxybenzoic acid derivatives with therapeutic potential, elucidate their mechanisms of action at the molecular level, and critically evaluate existing research to consolidate the current knowledge.

## 2. 4-Hydroxybenzoic Acid

4-HB or p-hydroxybenzoic acid is structurally characterized by a hydroxyl group (-OH) and a carboxylic acid group (-COOH) attached to a benzene ring at the para position [159].

This compound is particularly important for its role in the biosynthesis of coenzyme Q (CoQ), an essential component in cellular energy production, mitochondrial function, and metabolic homeostasis in bacteria, yeasts, and mammals [160]. As a precursor in this pathway, 4-HB contributes to the synthesis of CoQ, which is crucial for aerobic respiration in many organisms [160,161]. 4-HB biosynthesis begins in the cytosol, derived from tyrosine via both 4-hydroxyphenylpyruvate dioxygenase-like protein (HPDL)-dependent and HPDL-independent pathways [161,162]. Following its synthesis, 4-HB is transported into mitochondria, where it undergoes condensation with a polyprenoid tail catalyzed by the prenyltransferase COQ2 (Figure 2). Subsequent enzymatic modifications of the quinone ring are mediated by multiple enzymes constituting the CoQ biosynthetic complex [161]. This functional property links 4-HB to significant metabolic and physiological processes (Table 1).

### 2.1. Anti-Inflammatory Activity

The body’s natural response against microbial infection or cellular damage is mediated, among other elements, by activation of the NLRP3 inflammasome. This multiprotein complex is triggered by danger signals such as mitochondrial reactive oxygen species (ROS), microbial toxins, or cellular stress. Upon activation, NLRP3 recruits the adaptor protein ASC and procaspase-1, leading to the formation of the inflammasome complex. This results in the activation of caspase-1 and the maturation and release of the proinflammatory cytokines IL-1β and IL-18 [163].

4-HB exerts an anti-inflammatory effect by directly inhibiting the NLRP3 inflammasome, thus modulating inflammation. It suppresses NLRP3 inflammasome activation by decreasing ROS production, thanks to its intrinsic antioxidant properties. Since mitochondrial ROS are key activators of NLRP3, their reduction prevents inflammasome activation and contributes to an anti-inflammatory effect [13].

Additionally, 4-HB inhibits the release of IL-1β, contributing to the reduction of the inflammatory response. The inhibition of the Asc protein aggregation prevents the formation of the inflammasome complex [164]. It has been shown that 4-HB is capable of inhibiting this aggregation, thereby reducing the formation of the inflammasome complex [13]. Notably, 4-HB restricts NLRP3 inflammasome activation even without being converted into CoQ_10_, indicating that its anti-inflammatory effect is independent of its metabolic biotransformation [13].

At the transcriptional level, 4-HB modulates the expression of key inflammatory genes, reducing the transcription of IL-1β, TNF-α, IL-6, Nlrp3, and Casp1. It also regulates the Myd88 signaling pathway, suppressing IL-1β transcription even in the presence of an inhibitor of this pathway [13]. This suggests that 4-HB directly impacts the regulation of inflammation at the genetic level. At the cytokine level, 4-HB treatment significantly reduced mRNA and protein levels of pro-inflammatory cytokines such as IL-4, IL-6, and TNF-α. Additionally, 4-HB increased the mRNA and serum levels of IL-10, an anti-inflammatory cytokine, reinforcing its role in promoting a more balanced immune response [14,15]. 4-HB also reduces LPS-induced systemic inflammation by lowering IL-1β levels. This effect highlights 4-HB’s ability to mitigate widespread inflammation, which is Important in the context of systemic inflammatory disorders [13].

Inflammatory cells are essential for immune defense and inflammation regulation. Th17 and Treg cells, in particular, play a key role in balancing immune activation and suppression. Regarding inflammatory cells, 4-HB reduced the proportion of Th17 and Treg cells, suggesting an improvement in the inflammatory response by modulating the activity of these cells, which are critical in autoimmune and chronic inflammatory diseases [14].

### 2.2. Antioxidant Activity

The antioxidant capacity of 4-hydroxybenzoic acid is primarily attributed to its ability to neutralize free radicals and reduce oxidative stress, thereby protecting cells from damage. By scavenging reactive oxygen species and reactive nitrogen species (RNS), 4-HB helps prevent cellular damage to components such as lipids, proteins, and DNA. This antioxidant property is crucial in preventing the progression of chronic diseases associated with oxidative stress [16].

In addition, 4-HB is an active component in various plant extracts, such as *Ageratina ringens*. 4-HB has demonstrated significant antioxidant effects and cardiovascular health benefits. Its antioxidant activity is linked to its ability to scavenge free radicals, reducing oxidative damage. 4-HB also shows therapeutic potential in various conditions, including those related to hypertension [16].

In the intestinal context, 4-HB plays a protective role by enhancing the intestinal mucus barrier, increasing mucin production, and helping to prevent oxidative damage. This protective antioxidant effect helps maintain the integrity of the intestinal lining and prevent damage caused by oxidative stress. Additionally, 4-HB helps regulate mucin production, improving intestinal barrier function [14].

### 2.3. Therapeutic Applications

#### 2.3.1. Modulation of the Immune Response in Inflammatory Bowel Diseases

In the context of inflammatory bowel disease, 4-HB has been shown to be effective in reducing intestinal inflammation. It decreases the proportion of Th17 and Treg cells, two key cell types in the pathogenesis of these diseases. Additionally, 4-HB reduces the levels of the pro-inflammatory cytokines IL-4, IL-6, and TNF-α in the colon and serum, suggesting a significant reduction in the inflammatory response. Finally, 4-HB increases the levels of IL-10, an anti-inflammatory cytokine crucial for maintaining intestinal homeostasis and promoting the recovery of inflamed tissue [14,15].

#### 2.3.2. Anti-Inflammatory and Antitumor Effects

4-HB was identified as one of the key components in the profile of the Huanghua inverted water lotus. The 4-HB present in this plant has been linked to its anti-inflammatory and antitumor effects, as well as its influence on key metabolic pathways, including PI3K/Akt, MAPK3, and STAT3 signaling pathways [17].

#### 2.3.3. Antioxidant Role and Microbiome-Mediated Protection of the Intestinal Barrier

4-HB plays a crucial role in both antioxidant protection and modulation of the intestinal barrier. It enhances the intestinal mucus barrier by reversing the reduction in mucin expression, particularly MUC2, caused by DSS-induced colitis [14]. Additionally, 4-HB increases the number of goblet cells in the colon, promoting mucus production and protecting the intestinal mucosa from oxidative damage. These effects are likely linked to gut microbiome modulation, specifically by increasing *Akkermansia muciniphila*, a bacterium known to strengthen mucosal integrity and reduce oxidative stress. Moreover, 4-HB improves intestinal barrier function by regulating mucin production, restoring the expression of *Muc1*, *Muc2*, and *Muc3* [14].

#### 2.3.4. Metabolic Therapy for Primary CoQ Deficiency

Recently, 4-HB has gained attention in biomedical research for its therapeutic potential in disorders linked to CoQ biosynthesis deficiencies. 4-HB serves as a substrate for the COQ2 enzyme (Figure 2), and pathogenic variants in COQ2 may increase the enzyme’s Km, implying that substrate enhancement therapy could help. Studies show that 4-HB can restore CoQ levels in skin fibroblasts from patients with pathogenic variants in COQ2 [18,19].

In the *Coq2^A252V^* mouse model, which exhibits cardiac insufficiency, edema, and neurodevelopmental delay leading to perinatal lethality, treatment with 4-HB significantly improves mitochondrial function. The treatment increases the levels of CoQ_9_ and CoQ_10_, restoring the proper function of the mitochondrial respiratory chain. This enhancement helps rescue the mice from perinatal lethality and the multisystemic disease [19]. The results suggest that 4-HB holds therapeutic promise for addressing mitochondrial dysfunctions linked to COQ2 mutations, potentially offering a new avenue for treating related human diseases.

#### 2.3.5. Therapeutic Potential in Glucose Regulation and Diabetes

It has been shown that 4-HB enhances insulin secretion, modulates GLUT4 glucose transporter expression, and exhibits an antihyperglycemic effect through PPARγ activation, promoting glucose metabolism and glycogen storage in the liver. Additionally, in silico studies suggest that 4-HB acts as a dual agonist of PPARγ and GPR40 receptors, both critical targets in glucose homeostasis [20]. Furthermore, 4-HB has been shown to have a positive effect on weight loss in diet-induced obese mice, which is associated with a reduction in white adipose tissue percentage [35].

These findings position 4-HB as a promising candidate for the development of new antidiabetic treatments, improving insulin sensitivity and contributing to metabolic regulation.

#### 2.3.6. Neuroprotective Effect Inhibiting the Aggregation and Propagation of α-Synuclein

4-HB has demonstrated neuroprotective effects by inhibiting the aggregation and propagation of α-synuclein (αS), a key protein in the pathogenesis of neurodegenerative diseases such as Parkinson’s disease and multiple system atrophy (MSA) [21].

Studies in cellular models have shown that 4-HB reduces the formation of toxic αS aggregates, interfering with both its intracellular spread and cell-to-cell transmission. 4-HB inhibited αS seeding activity, limiting aggregate formation in the early stages of incubation. However, its protective effect diminished over time, possibly due to reduced cell viability or the inability of aggregate-containing cells to effectively transfer them to their daughter cells [21,22].

These findings suggest that 4-HB has potential therapeutic value in preventing αS-related neurodegeneration, although further research is needed to optimize its long-term effectiveness.

#### 2.3.7. Antimicrobial Effect

The 4-HB, present in maize root exudates, exhibits antimicrobial effects by interfering with *Phytophthora sojae* infection, the pathogen responsible for Phytophthora blight in soybean. This disease causes root rot, wilting, and can lead to plant death, especially in waterlogged soils. 4-HB inhibits zoospore motility and cystospore germination, contributing to disease suppression. This benefit is enhanced by the synergy with other phenolic acids, improving plants’ natural resistance to the pathogen. This interaction plays a key role in enhancing the effectiveness of intercropping systems in suppressing soil-borne diseases [23].

#### 2.3.8. Role in Antifungal Antibiotic Production

4-HB is a key regulator in the biosynthesis of the antifungal compound heat-stable antifungal factor (HSAF) in *Lysobacter enzymogenes*. Its production, dependent on the enzyme LenB2, is essential for maintaining adequate HSAF levels, a metabolite with potential for developing new antibiotics [24,165]. HSAF is crucial due to its ability to inhibit pathogenic fungi, offering a promising alternative against resistant fungal infections. Additionally, 4-HB interacts with the LysRLe regulator, enhancing the expression of HSAF biosynthetic genes and promoting its synthesis [24].

### 2.4. Pharmacokinetics and Toxicology

In vivo studies in rabbits showed that oral doses ranging from 100 to 1500 mg/kg of 4HB resulted in a total urinary recovery of 84% to 104%, indicating efficient absorption and rapid renal elimination. Metabolites conjugated with glucuronic acid (10–35%) and sulfate (4–7%) were identified in the urine, and all metabolite concentrations returned to baseline within 24 h. These findings suggest that 4-HBA has good oral bioavailability and is efficiently excreted via the urinary route [166].

4-Hydroxybenzoic acid also exhibits low acute toxicity in mammals, with oral and dermal LD50 values exceeding 2000 mg/kg in rats and rabbits, respectively, and an inhalation LC50 in rats greater than 0.47 mg/L (dust). In human research, 4-HBA has been administered orally without adverse effects: participants received up to 5 g every 6 h over 24 h, or a total of 26 g over 28 h, diluted in drinking water [167].

Regarding environmental toxicity, 4HB shows moderate to low toxicity toward aquatic organisms. The 48 h EC50 for *Daphnia magna* ranges from 173 to 1690 mg/L (median 932 mg/L), while the 96 h EC50 for algae is reported at 42.8 mg/L. Overall, these data suggest that 4HB poses limited toxic risk to terrestrial and aquatic organisms, although precaution is warranted in ecologically sensitive aquatic environments [168].

## 3. *ß*-Resorcylic Acid

BRA or 2,4-hydroxybenzoic acid, is a phytophenolic compound with two hydroxyl groups bound on the benzene group in the positions 2 and 4. It is a plant metabolite found in tea plants, robusta and arabica coffees, red wine, and fruits like cranberries, olives, or avocados [169,170,171].

### 3.1. Anti-Inflammatory Activity

BRA is a derivative of salicylic acid, the main metabolite of aspirin and the main responsible for its anti-inflammatory effect [29]. This structural relationship has focused the interest of many researchers trying to identify a possible anti-inflammatory effect also in BRA. In this way, BRA has shown a potent anti-neuroinflammatory effect in an animal model of mitochondrial disease due to a mutation in *Coq9*, decreasing CXCL10 and CCL2, two chemokines that exert critical roles in the migration and recruitment of leukocytes [25,26]. This neuroinflammatory activity is accompanied by a reduction in the expression of GFAP and microglial cells, as well as a change in the phenotype of the latter, from a proinflammatory and phagocytic form to a relaxed state [26,27]. However, this anti-inflammatory effect seems to be unrelated to a direct action on the inflammatory mechanism. Notably, BRA administration after NFKB induction with LPS administration is unable to decrease the release of the key proinflammatory cytokines such as IL-1β or TNFα; It also does not improve survival in models of neuroinflammation due to mitochondrial Complex I deficiency; nor does it exert its anti-inflammatory effects through mTORC1 inhibition, as evidenced by an unaltered S6RP/S6R ratio in the brain [27].

### 3.2. Antioxidant Activity

BRA acts as a free radical scavenger, capable of efficiently capturing a wide variety of free radicals via single-electron transfer [28]. The number and positioning of hydroxyl groups in the molecule enhance its reactivity, enabling it to reduce oxidative damage effectively [172]. However, it is the phenols with a single hydroxyl group in the ortho or para position that exhibit the best antioxidant properties [28].

### 3.3. Antimicrobial Activity

These compounds have been widely used in the food industry as flavorings and additives [171]. This property as a preservative is related to its potent antimicrobial capacity against Gram-negative bacteria (like *Escherichia coli*, *Pasteurella multocida*, or *Neisseria gonorrhoeae*) and Gram-positive (like *Staphylococcus aureus* and *Enterococcus faecalis*) [4,173].

### 3.4. Therapeutic Applications

#### 3.4.1. Neuroinflammation and Related Conditions

Neuroinflammation has been recognized as a key pathophysiological mechanism in various models of mitochondrial and neurodegenerative diseases, including encephalopathy associated with CoQ deficiency. The therapeutic applications of BRA in the context of mitochondrial and neurodegenerative diseases are of significant interest due to its potential to modulate both neuroinflammation and mitochondrial metabolism. Although the available evidence does not support a direct anti-inflammatory role for BRA, in the preclinical *Coq9^R239X^* model, which exhibits a fatal mitochondrial encephalopathy due to CoQ deficiency, treatment with this analog has shown significant anti-inflammatory benefits [26,27].

Specifically, BRA modulates the expression of inflammation-related genes, including *Bgn* (biglycan), *Ccl6* (chemokine C-C motif ligand 6), *Cst7* (cystatin F), *Ifi27l2a* (interferon alpha-inducible protein 27-like 2A), *Ifitm3* (interferon-induced transmembrane protein 3), *Itgax* (integrin alpha X), and *Vav1* [27]; normalized the plasma levels of proteins have been related to immune functions such as SERPINA, MASP1, and AI182371, and brain metabolites N-Ac-Glu and N-Ac-Glu-6P, which are related to neuroinflammatory and myelination processes [26].

Overall, although BRA’s primary effects are linked to CoQ metabolism and Q-junction function, its administration reduces reactive gliosis, astrogliosis, neuroinflammation, and spongiosis, thereby reversing the pathological phenotype in the encephalopathy model and significantly extending survival [26,27].

#### 3.4.2. Metabolic Therapy for CoQ Deficiency

A variety of studies, conducted by independent laboratories, have explored the therapeutic use of BRA for the treatment of primary coenzyme Q deficiencies. Specifically, BRA supplementation has demonstrated efficacy in primary deficiencies caused by mutations in the *Coq7* and *Coq9* genes [26,27,30,31,32]. In both cases, the administration of BRA, which contains the hydroxyl group introduced in the benzoquinone ring by the step catalyzed by COQ7-COQ9, will allow skipping the defective enzymatic reaction, thus achieving a bypass effect, restoring the biosynthetic pathway of CoQ [31,32] (Figure 2).

This bypass effect improves mitochondrial CoQ levels, reduces the accumulation of demethoxyubiquinone (DMQ), a toxic biosynthetic intermediate that builds up in *Coq7* and *Coq9* mutations, and stabilizes the Q complex [27,31,32]. These molecular effects are particularly pronounced in the heart, kidneys, liver, and skeletal muscle, resulting in a significant improvement in mitochondrial bioenergetics, especially in the liver, brain, and kidneys [27,31].

Moreover, the restoration of mitochondrial function is supported by BRA-induced normalization of the mitochondrial proteome and metabolic pathways. In the *Coq9^R239X^* preclinical model, BRA treatment corrects the expression of CoQ-dependent enzymes (e.g., ETFDH, PRODH2, DHODH, CHDH), as well as proteins involved in β-oxidation, folate and glycine metabolism, nucleotide biosynthesis, the tricarboxylic acid (TCA) cycle, the carnitine shuttle, and the oxidative phosphorylation (OxPhos) system [26].

Mutations in *Coq7* and *Coq9* typically result in mitochondrial encephalopathy due to CoQ deficiency. BRA treatment ameliorates mitochondrial dysfunction, leading to reduced spongiform degeneration, decreased reactive astrogliosis and oxidative stress, and ultimately the absence of brain lesions. Collectively, these therapeutic effects significantly improve disease phenotypes and prolong survival, surpassing the benefits observed with ubiquinol-10 supplementation [26,27,31].

Interestingly, BRA has also shown therapeutic potential in models with *Coq8a* and *Coq6* mutations. However, in these cases, the mechanism of action does not involve a bypass effect and remains poorly understood [33,34]. These findings suggest that BRA may exert beneficial effects through CoQ-independent mechanisms.

In *Coq8a^ΔPodocyte^* mice, BRA treatment prevented the onset of focal segmental glomerulosclerosis (FSGS) and podocyte foot process effacement, preserving renal function. This was accompanied by a reduced number of sclerotic glomeruli and improved plasma albumin levels, ultimately leading to significantly enhanced survival in this model [33]. Similarly, in the *Coq6^podKO^* mouse model of steroid-resistant nephrotic syndrome (SRNS), BRA administration prevented renal dysfunction and improved overall survival [34]. These results highlight BRA as a promising therapeutic approach for forms of human nephrotic syndrome stemming from primary defects in the CoQ_10_ biosynthesis pathway.

The modulatory effect of BRA on mitochondrial CoQ metabolism has also been proposed as a promising therapeutic strategy for the treatment of secondary CoQ deficiency [35]. Alterations in the CoQ biosynthetic pathway have been reported in the insulin-resistant white adipose tissue (WAT) of both mice and humans, resulting in secondary CoQ deficiency in this tissue and increased production of oxidative stress [36]. In this context, dietary supplementation with BRA in diet-induced obese (DIO) mice restores mitochondrial CoQ metabolism in WAT, reduces adipocyte hypertrophy, and counteracts associated metabolic abnormalities in WAT. These effects in WAT also contribute to the prevention of weight gain in BRA-treated animals [36].

#### 3.4.3. Therapeutic Effects on Diabetes and Metabolic Conditions

Insulin resistance is a hallmark in several metabolic diseases, including type II diabetes and obesity. The presence of different elements that disturb the homeostasis of glucose metabolism, like inflammation, hyperlipidemia, or diets high in fat and/or sucrose, may be the original cause of this insulin resistance [174,175,176]. In addition, different cellular stress situations related to mitochondria, like oxidative stress generation and mitochondrial dysfunction, have been described to play a causal role in this resistance [177]. The administration of BRA in an already obese mouse has been shown to induce a marked reduction in body weight, in a very effective manner [30,35]. This therapeutic effect is associated with metabolic remodeling in the liver, specifically through HFN4alfa/LXR-dependent transcriptomic activation of the hepatic lipid metabolism pathways, leading to enhanced lipid catabolism. Consequently, ectopic fat accumulation in the liver is prevented, thereby protecting against metabolic dysfunction-associated steatotic liver disease (MASLD). These hepatic effects of BRA supplementation, in conjunction with the normalization of mitochondrial CoQ metabolism in WAT (as described in Section 4.1), synergistically enhance glucose homeostasis and are primarily responsible for the substantial weight loss observed in DIO mice [35].

#### 3.4.4. Antitumor Effects

Dihydroxybenzoic acid-derived compounds have shown a direct effect as kinase inhibitors, which gives them excellent potential as an anticancer drug. In particular, BRA has shown a strong anticancer effect in vitro in different cancer cells by inhibiting cyclin-dependent kinase (CDK1) [37,38]. These cell cycle regulators, after binding to cyclin D, facilitate the progression through the different stages of the cell cycle, which makes them of great therapeutic interest for the treatment of different types of cancer, like human lung adenocarcinoma [178,179].

### 3.5. Pharmacokinetics and Toxicology

Not much information is available on the pharmacokinetic properties of BRA. In fact, a single article conducted in our laboratory determined that the plasma half-life of BRA after intravenous administration (50 mg/kg) is about 200 min, with the maximum concentration being reached five minutes after administration (1.63 × 10^5^ ng/mL). These results leave an area under the curve of 3.5 × 1.6 ng/mL/min, with a volume of distribution of 1.42 × 10^5^ (mg/kg)/(ng/mL) and a clearance (Cl) of 5.47 × 10^5^ (mg/kg)/(ng/mL). Accordingly, sustained administration of BRA would be necessary to achieve a therapeutically relevant plasma concentration [35].

Regarding toxicity, available data suggest that BRA is relatively well tolerated at moderate to high doses in animal models. In rats, oral administration of up to 6000 mg/day for 16 days in humans resulted primarily in glucuronide and sulfate conjugates in urine, indicating efficient phase II metabolism [180]. In reproductive toxicity studies, subcutaneous administration of up to 380 mg/kg to pregnant rats did not affect implantation rates, fetal weight, resorptions, or malformations [181]. However, a higher cumulative dose (428 mg/kg followed by 214 mg/kg) was associated with a reduction in serum calcium and a 33% rate of fetal toxicity, including minor skeletal malformations. These findings were not statistically analyzed, and the authors noted that the effects differed from those typically observed with salicylate exposure [182]. Overall, BRA appears to have a low acute toxicity profile, although potential developmental effects at high doses warrant further investigation.

## 4. Vanillic Acid

VA, or 4-hydroxy-3-methoxybenzoic acid, is a natural phenolic compound commonly used as a flavoring agent [183]. It represents an oxidized form of vanillin and is found in various plant sources, including the roots of *Angelica sinensis*, commonly known as “female ginseng” and largely used in traditional Chinese medicine, and the leaf extracts of *Ginkgo biloba* and *Poliomintha longiflora*, Mexican oregano [47]. VA has demonstrated remarkable therapeutic potential through its diverse mechanisms of action, particularly its anti-inflammatory, antioxidant, and regulatory effects on cellular processes (Table 1).

### 4.1. Anti-Inflammatory Activity

VA exerts potent anti-inflammatory effects by suppressing key inflammatory pathways. It inhibits the production of pro-inflammatory cytokines, including TNF-α, interleukin-6 (IL-6), and IL-1β. This suppression occurs primarily through the inhibition of NF-κB, a critical transcription factor for inflammatory gene expression [39,40,41,42]. In lipopolysaccharide-stimulated mouse peritoneal macrophages, VA effectively suppresses NF-κB activation, demonstrating its direct anti-inflammatory activity at the cellular level [40,184]. Additionally, in *Coq9^R239X^* mice, VA decreased the proinflammatory cytokines CXCL10 and CCL2 while the anti-inflammatory cytokines IL-6 and IL-10 showed a trend toward increased levels [26].

VA’s antioxidant properties play a significant role in its anti-inflammatory activity. It scavenges ROS and inhibits lipid peroxidation, thereby reducing oxidative damage and subsequent inflammation. This has been demonstrated in both cellular and animal models [39,56] and it will be described in detail later.

The inhibition of inflammatory enzymes represents another significant mechanism of VA’s anti-inflammatory activity. It downregulates cyclooxygenase-2 (COX-2) and inducible nitric oxide synthase (iNOS), crucial enzymes in the inflammatory cascade. Additionally, VA modulates mitogen-activated protein kinase (MAPK) and Janus kinase/signal transducer and activator of transcription (JAK/STAT) pathways, further contributing to its anti-inflammatory effects [40,41,42,43].

### 4.2. Antioxidant Activity

The antioxidant activity of VA, as well as that of other hydroxybenzoic acid derivatives, is significantly influenced by its chemical structure, particularly the presence of carboxyl and hydroxyl groups on the benzene ring that enhance its radical-scavenging capacity. Comparative studies have demonstrated that VA is more potent than vanillin due to these structural features [49].

Vanillic acid exhibits strong direct antioxidant effects by scavenging free radicals and reducing oxidative stress. Studies have demonstrated its ability to neutralize ROS, including hydroxyl radicals, superoxide anions, and peroxynitrite [39,48,49]. In in vitro and ex vivo models, VA acts as an effective hypochlorous acid and hydrogen peroxide scavenger, sometimes outperforming reference antioxidants like Trolox and aminoguanidine [39]. Its radical-scavenging capacity has been confirmed in various assays, including DPPH, ABTS, and ORAC, where it often surpasses the activity of ascorbic acid [48,50]. Interestingly, the analysis of the antioxidant properties of vanillic acid esters showed that their protective effect against free radical-induced biomembrane damage increases with greater lipophilicity, when compared to vanillic acid [50].

VA enhances the activity of endogenous antioxidant enzymes such as superoxide dismutase (SOD), glutathione peroxidase (GPx), and catalase (CAT). VA reduces oxidative stress in human umbilical vein endothelial cells by activating the AMPK signaling pathway, which enhances the expression of antioxidant enzymes [51]. In rat models, VA has been shown to protect against oxidative stress-induced cardiotoxicity by enhancing the activities of glutathione (GSH), CAT, GPx, and SOD [54]. Similar results were obtained in STZ-induced diabetic rats [52,53] and in Fe^2+^-induced rat brain damage [55].

VA effectively inhibits lipid peroxidation, a key marker of oxidative damage. In human plasma, VA prevented the formation of thiobarbituric acid-reactive substances (TBARS) and protein-bound carbonyls, demonstrating its ability to protect cellular membranes from oxidative damage [39,56]. In mice, VA reversed the increase in lipid peroxidation markers induced by sodium arsenite [56]. Similarly, in hamster models of oral carcinogenesis, VA significantly reduced lipid peroxidation levels [185].

### 4.3. Therapeutic Applications

#### 4.3.1. Neuroinflammation

In neuroinflammatory models, VA demonstrates significant neuroprotective effects by reducing glial cell activation and pro-inflammatory cytokine production [26]. Some studies have demonstrated that VA inhibits the NF-κB pathway, which is implicated in neurodegenerative diseases, including Alzheimer’s and Huntington’s [41,71,184]. Moreover, VA has demonstrated its anti-inflammatory effect over the encephalopathic phenotype of a CoQ deficiency model. In this context, VA normalizes the expression of canonical pathways related to inflammatory signaling analyzed by an RNA-Seq experiment on the mouse brainstem. Moreover, in the *Coq9^R239X^*, VA partially restores CoQ metabolism, particularly in the kidneys, and induces profound normalization of the mitochondrial proteome and metabolism, ultimately leading to reductions in gliosis, neuroinflammation, and spongiosis and, consequently, reversing the phenotype [26]. However, if the anti-inflammatory effects

#### 4.3.2. Gastrointestinal Inflammation

The anti-inflammatory properties of VA have shown promising results in treating gastrointestinal inflammatory disorders, particularly ulcerative colitis (UC). In experimental models, VA reduces colitis severity through the suppression of pro-inflammatory cytokines, preserving intestinal barrier integrity, and favorably modulating the gut microbiota [67,186]. Additionally, other studies have shown that VA inhibits ferroptosis, a form of cell death that exacerbates inflammation in the intestinal epithelium, providing a novel mechanism for its protective effects in colitis [44].

#### 4.3.3. Other Inflammatory Conditions

VA’s anti-inflammatory effect has also been tested in other pathological conditions, such as in the ovalbumin-induced asthma rat model, where VA reduces airway inflammation by inhibiting inflammatory mediator production and improving lung function. Its antioxidant properties also contribute significantly to these protective effects [45]. On the other hand, VA has been demonstrated to be beneficial to allergic inflammatory reactions in the human mast cell line (HMC-1). Treatment with VA significantly reduced pro-inflammatory cytokine levels, caspase-1 and NF-kB activity, as well as suppressed MAPK phosphorylation [46]. In musculoskeletal inflammation, particularly osteoarthritis models, VA reduces pain-related behavior and synovitis. It achieves this through inhibition of the NLRP3 inflammasome, a key mediator of inflammatory responses in synovial tissue [23]. Additionally, in murine inflammatory pain models, VA inhibits neutrophil recruitment, oxidative stress, and cytokine production alongside NF-kB activation, providing analgesic effects [47]. VA’s anti-inflammatory effects have also been described in other models of cardiovascular [43], renal [52,56] and hepatic inflammation [187].

#### 4.3.4. Cardiovascular Protection

VA has demonstrated remarkable cardioprotective effects by reducing oxidative damage. In H9c2 cardiomyocytes exposed to H_2_O_2_, VA was shown to protect against oxidative stress and cellular injury by reducing the levels of ROS, malondialdehyde (MDA), and lactate dehydrogenase (LDH), while improving the activity of antioxidant enzymes like SOD and glutathione. VA also increased the expression of key proteins involved in mitophagy, such as PINK1, Parkin, and Mfn2, and enhanced the LC3-II/LC3-I ratio while decreasing p62 levels [57]. Moreover, in the context of cardiac dysfunction, particularly cardiotoxicity and myocardial ischemia–reperfusion injury, VA improves mitochondrial function by reducing ROS production, enhancing antioxidant status, scavenging free radicals, and lowering lipid peroxidation levels [188].

#### 4.3.5. Metabolic Therapy for CoQ Deficiency

VA has shown potential in addressing CoQ deficiency as it is an analog of the natural precursor of the CoQ biosynthetic pathway, the 4-HB. The use of 4-HB analogs in CoQ deficiency is based on the idea that these compounds, already carrying the chemical modification catalyzed by the affected enzyme, can bypass defective steps in the biosynthetic pathway and restore CoQ production [12] (Figure 2). In this context, VA has shown promising effects in various models of CoQ deficiency due to mutations in *Coq6*. In *Saccharomyces Cerevisiae*, VA was able to rescue CoQ_6_ biosynthesis and respiration in a *Coq6* mutant [32,107,189], as well as restore growth in another *Coq6* yeast model [106]. In human *COQ6* mutant cells, VA restored CoQ_10_ levels, reduced ROS, and improved both cellular respiration and ATP production [190]. Nevertheless, VA can also alleviate the effects of CoQ deficiency caused by defects in proteins that are not directly bypassed by its action. In a *Drosophila* model with a mutation in *Coq2,* VA supplementation partially restored CoQ biosynthesis [58] and in human fibroblasts with mutations in *COQ9*, VA stimulated CoQ_10_ biosynthesis and improved viability [59]. However, VA had no impact on cells deficient in COQ4 or COQ7 [59]. In vivo, VA plays a crucial role in rescuing the encephalopathic phenotype in *Coq9^R239X^* mice. Its therapeutic effects appear to stem from two primary mechanisms: its influence on CoQ metabolism and mitochondrial function at the Q-junction, and its ability to mitigate neuroinflammation. In *Coq9^R239X^* mice, VA treatment did not modify CoQ levels, but it significantly decreased the DMQ/CoQ ratio in peripheral tissues. This tissue-specific response to VA induced an increase in mitochondrial bioenergetics. Moreover, VA produced a profound normalization of the mitochondrial proteome and metabolism, especially at the level of the Q-junction. Importantly, VA also induced anti-neuroinflammatory effects by the downregulation of proinflammatory pathways and diminishing the number and intensity of reactive astrocytes and microglia [26]. In this context, VA differs from BRA, the other 4-HB analog tested in *Coq9^R239X^* mice, in some aspects. For example, VA significantly upregulates COQ4, whereas β-RA has minimal impact on this protein. This effect of VA on COQ4 protein levels has also been demonstrated in vitro in fibroblasts with mutations in *COQ7* and *COQ9*, further supporting the idea that VA directly influences the regulation of this protein. This difference may help explain the distinct effects both compounds have on the DMQ/CoQ ratio [26,27,59].

#### 4.3.6. Therapeutic Effects on Diabetes and Metabolic Conditions

In diabetic models, VA reduces oxidative stress and improves insulin sensitivity. In a rat model of diabetic hypertension induced by a high-fat diet, treatment with VA for 8 weeks significantly reduced fasting glucose, insulin, and blood pressure levels. It significantly increased antioxidant activities (SOD, CAT, GPx, GSH, and vitamins C and E) while lowering lipid peroxidation markers [60]. Interestingly, VA derivatives isolated from the green algae *Cladophora socialis* have shown potent inhibitory activity against protein tyrosine phosphatase 1B (PTP1B), an enzyme involved in insulin cell signaling regulation [61]. The therapeutic potential of VA extends to metabolic syndrome and its components. VA activates the AMPK/Sirt1/PGC-1α pathway, modulates insulin signaling pathways such as Akt and ERK1/2, and regulates enzymes involved in glucose and lipid metabolism. In hyperlipidemia models, VA increases HDL-C levels while reducing serum cholesterol, triglycerides, free fatty acids, LDL-C, and VLDL-C. It also reduces HMG-CoA reductase activity and increases lecithin cholesterol acyl transferase activity, providing comprehensive lipid-modulating effects [62]. Moreover, VA reduces body weight in both high-fat diet-induced obesity and genetically obese mice. It suppresses adipogenesis by downregulating PPARγ and C/EBPα while activating AMPKα in white adipose and liver tissues. VA also inhibits lipid accumulation, lowers hepatotoxic and inflammatory markers, and reduces adipocyte differentiation. Additionally, VA enhances thermogenesis by increasing mitochondria-related factors in brown adipose tissue [63]. However, this effect on the obese phenotype remains controversial, as in another model of diet-induced obesity using a high-fat, high-sucrose diet, VA did not prevent body weight gain [35]. Nevertheless, VA demonstrates significant hepatoprotective effects in various models of liver injury [64,65]. In HepG2 cells under hyperinsulinemic conditions, VA mitigated mitochondrial dysfunction by reducing ROS and lipid peroxidation and activating the AMPK/Sirt1/PGC-1α pathway [66].

#### 4.3.7. Antitumor Effects

VA induces apoptosis in cancer cells by promoting mitochondrial apoptosis. Specifically, VA arrests the cell cycle at the G1 phase, inhibiting cancer cell proliferation. Moreover, it has been described that VA enhances the efficacy of chemotherapeutic drugs while minimizing adverse reactions. However, the VA anti-cancer mechanisms remain not fully understood [67]. In endometrial carcinoma models induced by N-methyl-N′-nitro-N-nitrosoguanidine (MNNG), VA treatment normalized histopathological alterations, reduced levels of thiobarbituric acid reactive substances (TBARS), lipid hydroperoxides, and cytochrome P450, while increasing antioxidant levels in plasma and uterus. It also downregulated the expression of MMP-2, MMP-9, and cyclin D1, with more pronounced effects in pre-treatment groups compared to co-treated subjects [68]. In a cancer-obesity comorbidity model, VA suppressed cancer growth by increasing apoptosis and autophagy markers. Mechanistic studies revealed that signal transducer and activator of transcription 3 (STAT3) phosphorylation was repressed by VA treatment, which was confirmed in a cell model of adipocyte conditioned medium-treated B16BL6 melanoma cells [69].

#### 4.3.8. Others

In cerebral hypoperfusion, a model based on transient bilateral common carotid artery occlusion in rats demonstrated that pretreatment with VA improved neurological deficits, reduced anxiety, attenuated reactive hyperemia, and protected the blood-brain barrier. This suggests its potential as a preventive agent in cerebrovascular disorders [70].

Additionally, VA has shown beneficial effects in osteoporosis by stimulating osteoblast proliferation and enhancing the expression of key markers involved in bone formation, which could help improve bone density and quality in rat osteoblast-like UMR 106 cells [72].

Moreover, VA exhibits potent antimicrobial activity against both Gram-positive and Gram-negative bacteria, inhibiting their growth, biofilm formation, and virulence. Notably, when combined with synthetic antibiotics, it enhances their effects against resistant pathogens, including those in the ESKAPE group (*Enterococcus faecium*, *Staphylococcus aureus*, *Klebsiella pneumoniae*, *Acinetobacter baumannii*, *Pseudomonas aeruginosa*, and *Enterobacter species*), positioning it as a promising option for combating difficult-to-treat infections [73].

### 4.4. Pharmacokinetics and Toxicology

A study using ultra-high-performance liquid chromatography tandem mass spectrometry (LC-MS/MS) demonstrated that after oral administration of vanillic acid at doses of 2, 5, and 10 mg/kg in rats, the plasma concentration peaks were observed at 0.42 ± 0.09, 0.73 ± 0.21, and 0.92 ± 0.28 μg/mL, respectively, within 0.55–0.64 h. The oral bioavailability was calculated to be between 25.3% and 36.2% [191]. The encapsulation of vanillic acid with hydroxypropyl-beta-cyclodextrin (HPBCD) significantly improved its solubility and bioavailability. The Tmax and AUC for the HPBCD-vanillic acid complex were 2 h and 253.46 ± 3.42, respectively, suggesting enhanced pharmacokinetic properties compared to the native form [192].

To date, no significant toxic effects have been reported for VA, except in some microbial models, where its antimicrobial properties can inhibit the growth of certain microorganisms. As mentioned in the text, VA and its isomers exhibit antimicrobial activity against a variety of pathogens. In terms of systemic toxicity, studies in Wistar rats have shown that oral administration of VA at a dose of 1000 mg/kg/day for two weeks does not produce significant alterations in hematological or biochemical parameters, suggesting a relatively safe toxicological profile at this dosage [193].

## 5. Protocatechuic Acid

Protocatechuic acid (PCA), chemically known as 3,4-dihydroxybenzoic acid, is a naturally occurring hydroxybenzoic acid widely distributed in plants, including berries, grapes, onions, and medicinal herbs like *Hibiscus sabdariffa* [103,194,195,196]. It is also found in olive oil and wine [194,197,198]. PCA often exists either in free form or conjugated form [196] and is a major microbial metabolite of anthocyanins and proanthocyanidins, contributing to its bioavailability [199,200]. Structurally, it features a catechol group, conferring strong antioxidant potential. PCA is synthesized via the shikimate/phenylpropanoid pathway [201] and has drawn interest for its multiple bioactivities, including antioxidant, anti-inflammatory, neuroprotective, and antidiabetic effects (Table 1).

### 5.1. Anti-Inflammatory Activity

PCA exerts potent anti-inflammatory effects through multiple molecular pathways. A central mechanism, shared with other 4-HB analogs such as syringic or gallic acids, is the inhibition of the NF-κB signaling pathway, a master regulator of inflammation [74]. In immune cells activated by inflammatory stimuli (e.g., bacterial lipopolysaccharide, LPS), PCA has been shown to prevent the activation of NF-κB. Specifically, PCA treatment blocks the degradation of IκB-α and phosphorylation of NF-κB p65, thereby retaining NF-κB in the cytosol and suppressing its transcriptional activity [75]. This leads to reduced expression of pro-inflammatory genes, including TNF-α and IL-1β, as shown both in vitro and in vivo [75,76]. Similar effects have been reported in other inflammatory disease models [77,78].

In parallel, PCA interferes with mitogen-activated protein kinase (MAPK) pathways that synergize with NF-κB in propagating inflammation. In LPS-challenged RAW264.7 macrophages, PCA inhibited the phosphorylation of key MAPKs (p38, c-Jun N-terminal kinase/JNK, and ERK1/2), further dampening inflammatory signal transduction [75]. Upstream of these pathways, PCA can target toll-like receptors—for example, in microglial cells, PCA was found to downregulate TLR4 expression and signaling, thereby blunting the entire cascade of TLR4-mediated NF-κB/MAPK activation [77]. Consistently, an in vitro study demonstrated that PCA inhibited Toll-like receptor 4-dependent NF-κB activation by suppressing upstream kinases Akt, mTOR, JNK, and p38 MAPK [79].

A direct consequence of NF-κB and MAPK inhibition is the downregulation of pro-inflammatory mediators. PCA reliably reduces the release of major cytokines and inflammatory enzymes in various models. In LPS-activated macrophages and microglia, PCA dose-dependently suppresses the secretion of TNF-α, IL-1β, IL-6, and the inflammatory lipid mediator PGE_2_ [75,77]. These effects may be tied to reduced expression of enzymes like COX-2 and iNOS, which produce PGE_2_ and nitric oxide (NO), respectively [75,80,81]. PCA’s inhibition of iNOS/NO production is particularly relevant in acute systemic inflammation—in an LPS-induced sepsis model, PCA administration markedly decreased plasma nitrite/nitrate (an indirect measure of NO) and reduced the surge of TNF-α (while also preventing an excessive IL-10 anti-inflammatory response) [76].

Beyond cytokines, PCA also interferes with the expression of cell adhesion molecules and chemokines that drive leukocyte recruitment to inflammatory sites. For example, PCA was shown to suppress VCAM-1 and ICAM-1 expression on TNF-α and 1β-stimulated endothelial cells [82,83]. These effects have been observed even at physiologically relevant concentrations, where PCA reduced soluble VCAM-1 secretion and VCAM-1 mRNA expression in vitro [84]. In vivo, this translates to reduced leukocyte infiltration: in hyperlipidemic ApoE^−^/^−^ mice, long-term dietary PCA supplementation lowered vascular ICAM-1/VCAM-1 expression and decreased monocyte–macrophage accumulation in atherosclerotic lesions [82]. Similarly, PCA downregulated the chemokine receptor CCR2 on monocytes, impairing their migratory response [82,85]. By blocking these adhesion and chemotactic signals, PCA mitigates the cellular component of inflammation (extravasation of neutrophils, monocytes, etc.), which is evidenced by reduced tissue leukocyte counts and exudation in PCA-treated animals [75,202].

Another mechanism contributing to PCA’s anti-inflammatory activity is the activation of endogenous regulatory pathways. PCA has been identified as a SIRT1 activator, promoting the deacetylation and inhibition of NF-κB, along with the downregulation of its upstream kinase IKKβ [86,87]. In vivo, this was associated with decreased levels of pro-inflammatory markers and increased expression of the anti-inflammatory factor PPARγ [87]. Additionally, PCA has been reported to induce HO-1 (heme oxygenase-1) via Nrf2 [88,89,90]. HO-1 is an anti-inflammatory, cytoprotective enzyme that can catabolize pro-oxidant heme and generate anti-inflammatory byproducts like bilirubin and carbon monoxide. By inducing HO-1, PCA further contributes to an anti-inflammatory environment.

In summary, PCA’s anti-inflammatory actions involve blocking pro-inflammatory signaling (TLR4–IKK–NFκB/MAPK pathways), thereby reducing the production of cytokines (TNF-α, IL-1β, IL-6, etc.), eicosanoids (PGE_2_ via COX-2), and reactive nitrogen species (NO via iNOS). Concurrently, it inhibits endothelial activation to limit leukocyte recruitment and augments endogenous anti-inflammatory and cytoprotective pathways (such as SIRT1/PPARγ and Nrf2/HO-1). These mechanistic insights provide a basis for PCA’s observed efficacy in diverse inflammation-driven pathologies, as discussed below.

### 5.2. Antioxidant Activity

The antioxidant potential of PCA is closely linked to its chemical structure and its ability to influence cellular redox balance through multiple mechanisms. Like other hydroxybenzoic acids, PCA acts as a direct free radical scavenger, modulates the expression and activity of endogenous antioxidant enzymes, and prevents lipid peroxidation, thereby offering protection against oxidative damage in various biological contexts [94,96,102].

PCA exerts strong direct antioxidant activity due to its catechol structure, which allows efficient hydrogen or electron donation to neutralize ROS such as superoxide anion, hydroxyl radical, and peroxyl radicals [94]. In chemical assays like 2,2-diphenyl-1-picrylhydrazyl (DPPH) and Oxygen Radical Absorbance Capacity (ORAC), PCA has shown low EC_50_ values, reflecting potent scavenging activity [102]. Its ability to stabilize the resulting semiquinone radical through resonance structures contributes to its antioxidant capacity [94,102]. Moreover, PCA demonstrates radical-scavenging effects in both aqueous and lipid environments, enhancing its protective effect on diverse cellular components [94]. Furthermore, PCA’s structure enables it to chelate transition metal ions (such as Fe^2+^ and Cu^2+^) that catalyze Fenton reactions, thereby indirectly preventing hydroxyl radical formation [95], though this chelation aspect is less studied compared to its radical-scavenging. Through these direct chemical interactions, PCA intercepts free radicals before they can damage lipids, proteins, or DNA.

In addition to its direct effects, PCA modulates endogenous antioxidant defenses by activating the Nrf2 pathway, a key transcription factor that controls the expression of numerous cytoprotective genes. Through Nrf2 activation, PCA upregulates antioxidant enzymes such as HO-1, SOD, CAT and GPx [89,90,96,97,98]. It also helps maintain adequate levels of reduced GSH, a major intracellular antioxidant [97,99]. These effects collectively improve the redox status of cells and enhance resilience against oxidative challenges in various disease models.

Lipid peroxidation is a key mechanism of cellular injury under oxidative stress, particularly affecting membranes and lipoproteins. PCA has demonstrated protective effects against lipid peroxidation by scavenging lipid peroxyl radicals and stabilizing membrane structures [102]. This action is reflected in reduced levels of lipid peroxidation markers such as malondialdehyde (MDA) and thiobarbituric acid reactive substances (TBARS) in tissues and plasma of treated animals [86,87,96,100,101]. PCA’s ability to preserve membrane integrity is especially relevant in pathological conditions such as ischemia-reperfusion injury, neurodegeneration, and cardiovascular disease, where oxidative damage to lipids plays a central role [96,98,203].

### 5.3. Therapeutic Applications

#### 5.3.1. Neuroinflammation

Chronic neuroinflammation, characterized by overactivation of glial cells (microglia and astrocytes) and release of neurotoxic inflammatory mediators, is a key contributor to neurodegenerative diseases. PCA has shown promising neuroprotective anti-inflammatory effects by modulating glial activation. In LPS-stimulated microglial cultures, PCA significantly inhibited the M1 (pro-inflammatory) activation state of microglia: treated cells released much lower levels of TNF-α, IL-1β, and NO, in conjunction with the suppression of NF-κB and MAPK signaling, as described earlier [91]. This translates in vivo to reduced neuroinflammatory damage. In a mouse model of global cerebral ischemia (which triggers robust inflammation in the brain), PCA administration attenuated the activation of microglia and astrocytes in the hippocampus and preserved blood–brain barrier integrity [93]. PCA-treated ischemic brains showed fewer degenerating neurons and less oxidative damage, suggesting that suppressing glial-driven inflammation helped protect neurons [93]. Moreover, PCA can influence microglial polarization: studies indicate that PCA pushes microglia toward an anti-inflammatory phenotype. In a model of intracerebral hemorrhage, PCA treatment suppressed the M1-type activation of microglia (characterized by production of TNF-α, IL-6, etc.) and was suggested to promote a shift toward the M2 (repair-oriented) state [92]. Indeed, in various neuroinflammatory models, PCA has been observed to lower CNS levels of cytokines and oxidative markers, correlating with reduced neurodegeneration [77,96,204,205].

#### 5.3.2. Hepatic Protection

Numerous studies indicate that PCA confers significant antioxidant protection in the liver. In a rat model of LPS-induced hepatic injury, PCA reduced inflammatory cell infiltration, congestion, and liver cell swelling, alleviating histopathological damage [103]. Likewise, its isopropyl ester derivative decreased hepatic malondialdehyde levels in mice challenged with LPS plus D-galactosamine (GalN), highlighting PCA’s ability to curb lipid peroxidation in vivo [76]. Further in vitro evidence shows that PCA protects hepatocytes from hydrogen peroxide (H_2_O_2_)-triggered oxidative stress, as it mitigates ROS formation and endoplasmic reticulum (ER) stress [104]. Furthermore, PCA beneficially modulated oxidative stress markers in murine models of NAFLD by reducing TBARS and improving lipid profiles [105], an effect partly attributed to NRF2 pathway activation and ferroptosis inhibition [206]. PCA also limited oxidative damage triggered by copper oxide nanoparticles through restoration of glutathione-based defenses in liver cells [99]. Notably, co-administration of PCA during chemotherapy lessened vincristine-induced oxidative stress in hepatic tissue, indicating that PCA’s antioxidant potency extends to toxin-induced injuries as well [207]. Together, these findings underscore PCA’s multifaceted antioxidant capacity in various hepatic injury models.

#### 5.3.3. Cardiovascular Protection

Although short-term red wine consumption containing PCA did not acutely alter lipoprotein oxidizability ex vivo [198], other experiments underscore PCA’s antioxidant potential in cardiovascular settings. PCA derived from extra virgin olive oil impeded the cell-mediated oxidation of LDL, maintaining intracellular redox balance and restoring activities of glutathione-related enzymes [197]. Similarly, studies of phenolic metabolites, including PCA, revealed that they reduce VCAM-1 secretion in endothelial cells, an early step in atherogenesis [84]. Moreover, in apolipoprotein E-deficient mice, supplementation with PCA suppressed monocyte adhesion, decreased NF-κB activity, and ultimately restrained atherosclerotic lesion development [82]. These findings suggest that PCA’s capacity to mitigate oxidative processes in the vasculature may help preserve endothelial function and curb inflammation-driven cardiovascular disorders.

#### 5.3.4. Metabolic Therapy for Primary CoQ Deficiency

PCA has recently emerged as a promising therapeutic candidate within the spectrum of primary coenzyme Q deficiencies. Among these, mutations in COQ6—encoding a flavin-dependent monooxygenase responsible for the C5-hydroxylation of the benzoquinone ring—lead to impaired CoQ biosynthesis and mitochondrial dysfunction (Figure 2). In yeast models lacking Coq6, this block results in the accumulation of prenylated intermediates such as 3-hexaprenyl-4-hydroxybenzoic acid, halting the biosynthetic process [107]. Remarkably, PCA, which already carries a hydroxyl group at position C5, bypasses this defective step. After prenylation by Coq2, the molecule is efficiently processed through the downstream enzymatic reactions, restoring Q6 production and respiratory growth in *S. cerevisiae* Δcoq6 strains in a dose-dependent manner, similar to VA [107].

The therapeutic potential of PCA extends beyond yeast knockout models. When human pathogenic COQ6 alleles were expressed in yeast, supplementation with PCA rescued oxidative growth across all hypomorphic variants tested [106]. These findings support its consideration as a bypass precursor with potential translational value, particularly in COQ6-related deficiencies where residual biosynthetic activity is retained.

### 5.4. Pharmacokinetics and Toxicology

A study using a validated and sensitive LC-MS/MS method demonstrated that after oral administration of protocatechuic acid at a dose of 50 mg/kg in mice, the plasma concentration peaked at 73.6 μM within 4.87 min. The compound showed rapid absorption with an initial half-life of 2.9 min and a terminal half-life of 16 min. PCA remained detectable in plasma for up to 8 h, and the area under the plasma concentration–time curve (AUC_0_→_8_h) was 1456 μM·min, indicating fast systemic exposure and two-phase elimination kinetics [208].

In the same study, PCA was also detected in human plasma at low ng/mL concentrations after oral intake of 60 g/day of black raspberry (BRB) powder for 21 days, reflecting its formation as a microbial metabolite of dietary anthocyanins [208]. This observation is further supported by other studies showing the presence of PCA and related metabolites in urine following oral administration of 500 mg of uniformly ^13^C-labeled cyanidin-3-glucoside to a cohort of human subjects [209].

Additional data from a separate pharmacokinetic study using Hedyotis diffusa Willd extract—a traditional medicinal plant naturally containing PCA among other phenolics—showed a distinct kinetic profile. Following oral administration of 4.837 g/kg of the extract in rats (providing a calculated PCA dose of 32.38 mg/kg), the plasma C_max of PCA was 119.95 ng/mL at 0.393 h, with a distribution half-life of 7.19 h and a markedly prolonged terminal half-life of 843.45 h. The AUC_0_→∞ was 10,177.69 ng·h/mL [210]. Although differences in dose, formulation, and biological matrix preclude direct comparison with the pure compound, these results suggest that the presence of other plant constituents may influence the pharmacokinetics of PCA, possibly by modulating its absorption or metabolism.

Regarding toxicity, PCA is generally considered safe at dietary levels, but high doses have been associated with mild hepatotoxic and nephrotoxic effects. In ICR mice, intraperitoneal administration of 500 mg/kg PCA significantly reduced glutathione (GSH) levels in the liver and kidney, and increased plasma ALT and urea levels, suggesting oxidative stress-related toxicity. Moreover, subchronic exposure via drinking water (0.1% for 60 days) led to sustained renal GSH depletion and elevated ALT activity, without major systemic toxicity [211].

## 6. Hydroxycinnamic Acids

Hydroxycinnamic acids have a C6-C3 structure, which includes a benzene ring (C6) attached to a three-carbon side chain (C3) that terminates in a carboxyl group [212]. This characteristic structure influences their biological activities, bioavailability, and applications in health and disease prevention (Table 1) [213].

The most prevalent hydroxycinnamic acids include *p*-coumaric acid, caffeic acid, ferulic acid, and sinapic acid [214].

### 6.1. p-Coumaric Acid

#### 6.1.1. Biological Activities

*p*-CA acid possesses strong antioxidant and anti-inflammatory properties, making it a promising compound for managing metabolic and inflammatory diseases [212,215].

*p*-CA has been shown to neutralize free radicals and diminish oxidative stress by enhancing hepatic fatty acid oxidation and increasing fecal lipid excretion. *p*-CA’s antioxidant properties help in preventing cardiovascular diseases [108]. It also modulates the electrical properties of biological membranes, which may affect cellular signaling [111].

Additionally, *p*-CA lowers the production of pro-inflammatory cytokines and adipokines associated with insulin resistance, thus offering benefits in metabolic disorders such as obesity and diabetes [113].

*P*-CA inhibits tumor growth by triggering apoptosis in cancer cells and suppressing angiogenesis. It also reduces the affinity of mutagenic compounds to DNA, helping to prevent mutations that could lead to cancer [114,116].

Likewise, using 13C-labeled compounds to trace incorporation into CoQ, *P*-coumarate has been shown to serve as a viable ring precursor in *E. coli*, *Saccharomyces cerevisiae* (yeast), and mammalian cells, suggesting an alternative pathway for 4HB production in these organisms [216].

#### 6.1.2. Therapeutic Applications

Despite its strong biological activities, *p*-CA’s bioavailability is relatively low, particularly in its conjugated forms. This limitation affects its therapeutic potential, and further research is needed to improve its absorption and stability [217].

### 6.2. Caffeic Acid

#### 6.2.1. Biological Activities

CA is a well-recognized antioxidant thoroughly studied for its anticancer, anti-inflammatory, and neuroprotective properties [119,212,218].

CA boosts the levels of antioxidant enzymes such as GPx and SOD, while also decreasing the production of ROS [119,120]. Moreover, CA and its derivatives, such as caffeic acid phenethyl ester (CAPE), are being explored for their anticancer properties, especially in breast and colon cancer [119]. Additionally, CA helps inhibit metastasis by suppressing epithelial-to-mesenchymal transition (EMT) and modulating critical signaling pathways such as PI3K/Akt and AMPK [119,133]. Notably, CA regulates microglial activation in the hippocampus, which may provide benefits in neurodegenerative diseases like Alzheimer’s [134]. Additionally, CA is used in skincare products due to its antioxidant and anti-aging properties [123].

#### 6.2.2. Therapeutic Applications

CA’s bioavailability is hindered by its rapid metabolism and excretion, and its pro-oxidant effects in the absence of tocopherols can be a drawback, necessitating careful formulation for therapeutic applications [119,219].

### 6.3. Ferulic Acid

#### 6.3.1. Biological Activities

FA is recognized for its anti-aging, anti-inflammatory, and neuroprotective effects [212].

FA effectively scavenges free radicals and boosts the cell stress response by upregulating cytoprotective systems. It also protects against UV-induced skin damage and carcinogenesis, making it a popular ingredient in cosmetics for its anti-aging and photoprotective properties [136]. Likewise, FA modulates inflammatory pathways by inhibiting pro-inflammatory cytokines production and regulating NF-κB and p38 MAPK signaling [141]. It also reduces platelet aggregation, lowering the risk of thrombosis and making it beneficial in preventing cardiovascular diseases [146]. FA shows potential in treating neurodegenerative diseases by alleviating oxidative stress and inflammation in the brain [220].

#### 6.3.2. Therapeutic Applications

Despite its wide-ranging therapeutic potential, FA’s low solubility and rapid metabolism limit its bioavailability. Nanoencapsulation and other delivery systems are being investigated to overcome these challenges [221].

### 6.4. Sinapic Acid

#### 6.4.1. Biological Activities

SA is a hydroxycinnamic acid known for its strong antioxidant, anticancer, and anti-inflammatory properties [212,222].

SA has been shown to protect lysosomes from dysfunction and mitigate oxidative damage by scavenging free radicals. It also enhances the activity of antioxidant enzymes [151]. Additionally, SA suppresses T-helper 2 immune responses, indicating its potential as a therapeutic agent for allergic asthma and other inflammatory conditions [154]. It also plays a role in modulating lipid metabolism, which may benefit cardiovascular health [116]. Likewise, SA promotes apoptosis in cancer cells by increasing caspase-3 activity and inhibiting cell invasion. SA’s anticancer properties are being actively explored, particularly in prostate and lung cancer [157].

#### 6.4.2. Therapeutic Applications

Similarly to the other hydroxycinnamic acids, SA’s bioavailability is restricted due to its rapid metabolism and low absorption in the gastrointestinal tract. While conjugates of SA show higher biological activity, their poor absorption presents a challenge for therapeutic applications [151].

### 6.5. Pharmacokinetics and Toxicology

Hydroxycinnamic acids are small phenolic acids abundant in plant foods. After oral intake, these compounds exhibit rapid but limited absorption, extensive first-pass metabolism, and efficient renal clearance. Absorption occurs via passive diffusion and likely via monocarboxylate transporters (e.g., MCT/SLC5A8); sinapic acid in particular is reported to use a Na^+^-dependent monocarboxylate transporter in the proximal gut [223]. These acids are absorbed very quickly: in general, their absorption half-lives are on the order of a few minutes (t_1/2 ≈ 0.07–0.08 h), and time-to-peak (T_max) is typically <1 h. Indeed, most is taken up before reaching the colon. Gastric uptake has also been observed, consistent with short T_max.

Once absorbed, hydroxycinnamic acids undergo extensive hepatic metabolism, especially by cytochrome P450s (CYPs), catechol-O-methyltransferase (COMT), UDP-glucuronosyltransferases (UGTs), and sulfotransferases (SULTs) [224].

The bioavailability of free hydroxycinnamic acids is relatively low due to first-pass metabolism. Plasma concentrations of unchanged acid are generally small [225].

They are relatively hydrophilic and do not concentrate heavily in fat or cross membranes extensively. Some tissue uptake may occur in rodents: quantitative studies reveal that caffeic acid distributes to the kidney, liver, muscle, lung, heart, spleen, and testis. Ferulic acid showed even wider tissue presence in the kidney, with smaller fractions in the liver, lung, heart, spleen, and brain [224]. On the other hand, quantitative tissue distribution data in humans are scarce.

After metabolism, elimination is mainly renal. Conjugated metabolites (glucuronides, sulfates) of the hydroxycinnamates are water-soluble and excreted in urine. In a human study, only ~11% of an ingested caffeic acid dose appeared as unchanged caffeic acid in urine; the remainder presumably was excreted as conjugates or oxidized forms over time [226]. For ferulic and coumaric acids, analogous urinary excretion of conjugates is expected. A portion of intake may also be eliminated via bile into feces; however, since free absorption is high, fecal loss of parent acid is minimal. Overall, elimination half-lives in humans appear generally short (≤1–2 h) for free forms, with much of the dose rapidly conjugated [225].

Acute and chronic toxicity studies indicate that hydroxycinnamic acids have low inherent toxicity at dietary or even nutraceutical doses. In human trials, very high supplementation has been well tolerated. For instance, 1000 mg/day of ferulic acid for six weeks led to substantial pharmacodynamic effects (improved lipid profiles) and caused no reported adverse effects and no changes in liver or kidney function tests [227]. Although this indicates a favorable safety profile and systemic exposure, formal toxicological limits in humans have yet to be established.

Overall, no significant safety concerns have been noted at realistic intake levels. These acids are generally recognized as safe in foods.

## 7. Conclusions and Perspectives

The hydroxybenzoic and hydroxycinnamic acid derivatives reviewed exhibit a wide range of therapeutic properties through different mechanisms of action that converge in a synergistic manner. These include anti-inflammatory and antioxidant effects, as well as their metabolic effects on the CoQ synthesis pathway. These compounds primarily exert their anti-inflammatory actions by inhibiting critical signaling pathways such as NF-κB, MAPK, and the NLRP3 inflammasome, thereby reducing the production of pro-inflammatory cytokines, chemokines, and enzymes like COX-2 and iNOS. Additionally, they help restore immune balance by modulating immune cell responses and promoting the release of anti-inflammatory mediators. This effect on different inflammatory pathways makes them especially interesting in inflammatory digestive pathologies and neuroinflammatory processes.

Another important mechanism of action is its antioxidant effect, which is facilitated through both direct scavenging of reactive species and the enhancement of the body’s intrinsic antioxidant defenses. By neutralizing free radicals and activating protective cellular pathways like Nrf2, these compounds prevent oxidative damage to key biomolecules, including lipids, proteins, and DNA. They also upregulate antioxidant enzymes, such as SOD, CAT, and GPx, while stabilizing redox balance and inhibiting lipid peroxidation, which helps maintain cellular integrity under oxidative stress conditions. This action is of vital importance in pathologies such as cancer, where the production of free radicals contributes significantly to the severity of the pathology.

Beyond their anti-inflammatory and antioxidant capacities, hydroxybenzoic acid derivatives offer a broad spectrum of therapeutic potential. Their ability to modulate key molecular pathways involved in mitochondrial function, immune regulation, and metabolic homeostasis positions them as promising candidates for addressing a variety of chronic conditions, including inflammatory diseases, neurodegenerative disorders, metabolic syndromes, and certain forms of cancer. Of relevance is their role in improving mitochondrial function by stimulating the endogenous synthesis of CoQ, acting both as a bypass mechanism and with other effects. This role in improving CoQ synthesis has positioned them as excellent therapeutic tools for the treatment of primary CoQ deficiencies due to mutations in some of the enzymes involved in the synthesis of this lipid. Thus, for example, 4HB has shown exceptional pharmacological effects, being able to rescue perinatal death, in a model of primary deficiency due to mutations in Coq2; or BRA and VA, which enhance CoQ synthesis in mutations in Coq7 or Coq9, for example. In addition, the fact that secondary deficiencies have also been described in pathologies of high incidence, such as obesity, positions them as compounds of great interest to the pharmaceutical industry.

From a translational perspective, these findings open several promising avenues for future research. One critical next step would be the optimization of formulation and delivery strategies to improve the bioavailability and tissue targeting of these compounds. Additionally, more comprehensive preclinical and clinical studies are needed to validate their efficacy and safety in humans across various disease models. Given their multitargeted mechanisms of action, the development of combinatorial therapies that incorporate hydroxybenzoic or hydroxycinnamic acid derivatives with conventional treatments may also enhance therapeutic outcomes. Finally, their natural origin and favorable safety profile make them attractive candidates for inclusion in preventive strategies and nutraceutical formulations aimed at reducing the risk or delaying the onset of chronic diseases associated with inflammation, oxidative stress, and mitochondrial dysfunction.

## Figures and Tables

**Figure 1 antioxidants-14-00711-f001:**
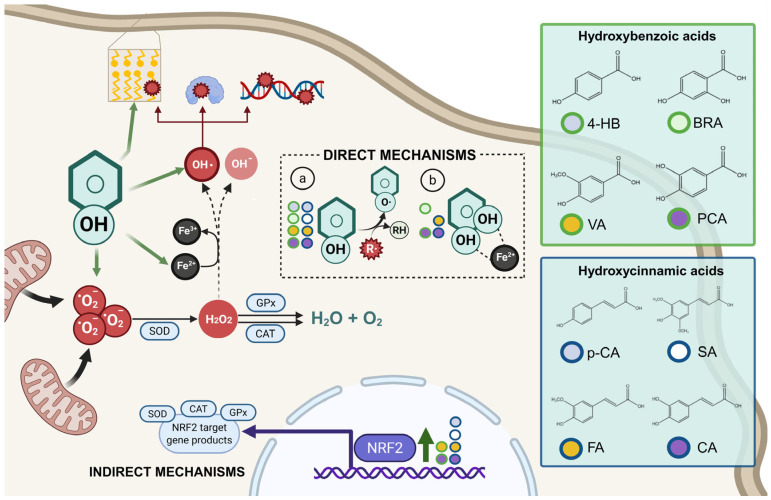
Antioxidant mechanisms of hydroxybenzoic and hydroxycinnamic acids. Phenolic compounds exert antioxidant effects through direct and indirect mechanisms. In direct mechanisms: (**a**) they neutralize free radicals (R·) by donating a hydrogen atom from the phenolic hydroxyl group (Ar–OH), resulting in a stabilized phenoxyl radical (Ar–O·) and a non-radical product (R–H); (**b**) some phenolic compounds can chelate transition metal ions such as Fe^2+^, thereby reducing the catalytic formation of highly reactive species via Fenton reactions. Indirect mechanisms involve the activation of the NRF2 signaling pathway, which upregulates antioxidant enzymes such as superoxide dismutase (SOD), catalase (CAT), and glutathione peroxidase (GPx). SOD catalyzes the dismutation of superoxide anion (•O_2_^−^) into hydrogen peroxide (H_2_O_2_), which is further detoxified by CAT and GPx into water (H_2_O) and oxygen (O_2_). If not neutralized, H_2_O_2_ can participate in Fenton reactions, generating hydroxyl radicals (•OH), extremely reactive species capable of inducing lipid peroxidation, protein oxidation, and DNA damage, thus amplifying cellular oxidative injury.

**Figure 2 antioxidants-14-00711-f002:**
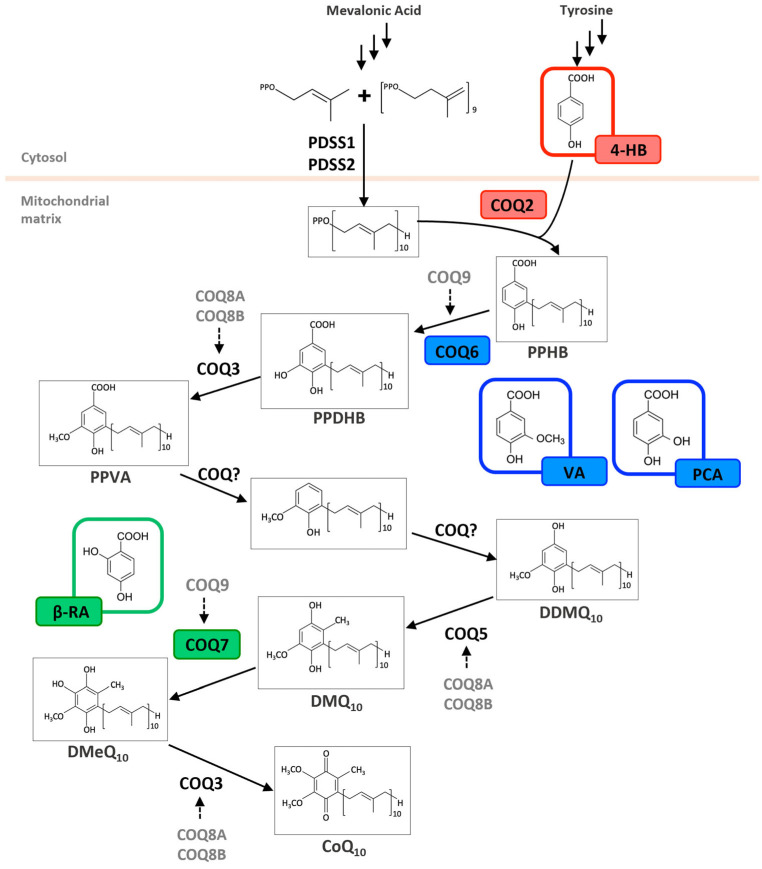
Metabolic effects of hydroxybenzoic acid derivatives on the coenzyme Q biosynthesis pathway. Schematic biosynthetic pathway of Coenzyme Q_10_ (CoQ_10_) in humans, starting from precursors derived from the mevalonate pathway and tyrosine metabolism. The polyprenyl side chain is synthesized by the PDSS1/PDSS2 complex and attached to 4-hydroxybenzoic acid (4-HB) by COQ2. Subsequent ring modifications involve a series of enzymes (COQ3–COQ9), resulting in the formation of CoQ_10_. Alternative ring substrates—vanillic acid (VA), protocatechuic acid (PCA), and β-resorcylic acid (β-RA)—are highlighted in colored boxes and shown entering the pathway at positions where they are hypothesized to bypass. PPHB = polyprenyl-4-hydroxybenzoate; PPDHB = Polyprenyl-3,4-dihydroxybenzoate; PPVA = Polyprenyl-4-hydroxy-3-methoxybenzoate; DDMQ_10_ = demethoxy-demethyl-ubiquinone-10; DMQ_10_ = demethoxyubiquinone-10; DMeQ_10_ = demethylubiquinone-10.

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
