# Peer review of "Natural Hydroxybenzoic and Hydroxycinnamic Acids Derivatives: Mechanisms of Action and Therapeutic Applications"

_antioxidants, 2025, doi:10.3390/antiox14060711_

Round 1
Reviewer 1 Report
This paper reviews available data on hydroxybenzoic and hydroxycinnamic acids derivatives. It combines different approaches – chemistry, biochemistry and pharmacology – and points out possible therapeutic applications, based on the cellular and molecular mechanisms of these molecules. Well written and well referenced, it deserves publication.
Nevertheless, I have some comments and criticisms, which should help to improve the paper.
- Only in vitro and preclinical data are presented; are they some clinical data available? This should be discussed.
- When available, pharmacokinetic data should be mentioned for all molecules discussed in this review. Toxicology of the substances should be discussed.
- Some perspectives should be given.
Some minor criticisms to improve the readability of the text.
- L 178: the sentence “4-HB is an active component in various plant extracts, such as Ageratina ringens, WHERE it contributes to both antioxidant effects and cardiovascular health benefits.” should be corrected.
- L 307,…: microorganism names should be in italic as before. In vitro, in vivo, ex vivo should be in italic all along the text.
- The chemical structures of all the substances that are discussed in this review should be presented in a separate figure.
See above.
Author Response
|
1. Summary |
|
|
|
We would like to thank the reviewer for the positive comments and for the interest in this review. We also appreciate the useful recommendations to improve the manuscript.
|
||
|
2. Point-by-point response to Comments and Suggestions for Authors |
||
|
Comments 1: Only in vitro and preclinical data are presented; are they some clinical data available? This should be discussed. |
||
|
Response 1: We thank the reviewer for this valuable comment. As correctly pointed out, most of the available evidence regarding hydroxybenzoic and hydroxycinnamic acid derivatives comes from in vitro and preclinical studies, which form the core of our review. Nevertheless, there is still a lack of well-controlled, compound-specific clinical trials. We have now clearly stated this limitation in the discussion of the manuscript and highlighted it as a key direction for future research: “Additionally, more comprehensive preclinical and clinical studies are needed to validate their efficacy and safety in humans across various disease models.” This change can be found on page 31 and on line 1168.
|
||
|
Comments 2: When available, pharmacokinetic data should be mentioned for all molecules discussed in this review. Toxicology of the substances should be discussed. |
||
|
Response 2: We thank the reviewer for this important observation. In response, we have now included a specific subsection for each compound within the manuscript, summarizing the available pharmacokinetic and toxicological data for the compounds discussed. The new paragraphs read as follows:
2.4. Pharmacokinetic and toxicology In vivo studies in rabbits showed that oral doses ranging from 100 to 1500 mg/kg of 4HB resulted in a total urinary recovery of 84% to 104%, indicating efficient absorption and rapid renal elimination. Metabolites conjugated with glucuronic acid (10–35%) and sulfate (4–7%) were identified in the urine, and all metabolite concentrations returned to baseline within 24 hours. These findings suggest that 4-HBA has good oral bioavailability and is efficiently excreted via the urinary route [164]. 4-Hydroxybenzoic acid also exhibits low acute toxicity in mammals, with oral and dermal LD50 values exceeding 2000 mg/kg in rats and rabbits, respectively, and an inhalation LC50 in rats greater than 0.47 mg/L (dust). In human research, 4-HBA has been administered orally without adverse effects: participants received up to 5 grams every 6 hours over 24 hours, or a total of 26 grams over 28 hours, diluted in drinking water [165]. Regarding environmental toxicity, 4HB shows moderate to low toxicity toward aquatic organisms. The 48-hour EC50 for Daphnia magna ranges from 173 to 1690 mg/L (median 932 mg/L), while the 96-hour EC50 for algae is reported at 42.8 mg/L. Overall, these data suggest that 4HB poses limited toxic risk to terrestrial and aquatic organisms, although precaution is warranted in ecologically sensitive aquatic environments [166]. This change can be found on page 15 and on line 293.
3.5. Pharmacokinetic and toxicology Not much information is available on the pharmacokinetic properties of BRA. In fact, a single article conducted in our laboratory determined that the plasma half-life of BRA after intravenous administration (50 mg/kg) is about 200 minutes, with the maximum concentration being reached five minutes after administration (1.63x105 ng/ml). These results leave an area under the curve of 3.5x1'6 ng/ml/min, with a volume of distribution of 1.42 × 105 (mg/kg)/(ng/ml) and a clearance (Cl) of 5.47 × 105 (mg/kg)/(ng/ml). Accordingly, sustained administration of BRA would be necessary to achieve a therapeutically relevant plasma concentration [35]. Regarding toxicity, available data suggest that BRA is relatively well tolerated at moderate to high doses in animal models. In rats, oral administration of up to 6000 mg/day for 16 days in humans resulted primarily in glucuronide and sulfate conjugates in urine, indicating efficient phase II metabolism [180]. In reproductive toxicity studies, subcutaneous administration of up to 380 mg/kg to pregnant rats did not affect implantation rates, fetal weight, resorptions, or malformations [181]. However, a higher cumulative dose (428 mg/kg followed by 214 mg/kg) was associated with a reduction in serum calcium and a 33% rate of fetal toxicity, including minor skeletal malformations. These findings were not statistically analyzed, and the authors noted that the effects differed from those typically observed with salicylate exposure [182]. Overall, BRA appears to have a low acute toxicity profile, although potential developmental effects at high doses warrant further investigation. This change can be found on page 18 and on line 454.
4.4. Pharmacokinetic and toxicology A study using ultra-high-performance liquid chromatography tandem mass spectrometry (LC-MS/MS) demonstrated that after oral administration of vanillic acid at doses of 2, 5, and 10 mg/kg in rats, the plasma concentration peaks were observed at 0.42 ± 0.09, 0.73 ± 0.21, and 0.92 ± 0.28 μg/ml, respectively, within 0.55-0.64 hours. The oral bioavailability was calculated to be between 25.3% and 36.2% [191]. The encapsulation of vanillic acid with hydroxypropyl-beta-cyclodextrin (HPBCD) significantly improved its solubility and bioavailability. The Tmax and AUC for the HPBCD-vanillic acid complex were 2 hours and 253.46 ± 3.42, respectively, suggesting enhanced pharmacokinetic properties compared to the native form [192]. To date, no significant toxic effects have been reported for VA, except in some microbial models, where its antimicrobial properties can inhibit the growth of certain microorganisms. As mentioned in the text, VA and its isomers exhibit antimicrobial activity against a variety of pathogens. In terms of systemic toxicity, studies in Wistar rats have shown that oral administration of VA at a dose of 1000 mg/kg/day for two weeks does not produce significant alterations in hematological or biochemical parameters, suggesting a relatively safe toxicological profile at this dosage [193]. This change can be found on page 23 and on line 687.
5.4. Pharmacokinetic and toxicology A study using a validated and sensitive LC-MS/MS method demonstrated that after oral administration of protocatechuic acid at a dose of 50 mg/kg in mice, the plasma concentration peaked at 73.6 μM within 4.87 minutes. The compound showed rapid absorption with an initial half-life of 2.9 minutes and a terminal half-life of 16 minutes. PCA remained detectable in plasma for up to 8 hours, and the area under the plasma concentration–time curve (AUC₀→₈h) was 1456 μM·min, indicating fast systemic exposure and two-phase elimination kinetics [208]. In the same study, PCA was also detected in human plasma at low ng/mL concentrations after oral intake of 60 g/day of black raspberry (BRB) powder for 21 days, reflecting its formation as a microbial metabolite of dietary anthocyanins [208]. This observation is further supported by other studies showing the presence of PCA and related metabolites in urine following oral administration of 500 mg of uniformly ¹³C-labeled cyanidin-3-glucoside to a cohort of human subjects [209]. Additional data from a separate pharmacokinetic study using Hedyotis diffusa Willd extract—a traditional medicinal plant naturally containing PCA among other phenolics—showed a distinct kinetic profile. Following oral administration of 4.837 g/kg of the extract in rats (providing a calculated PCA dose of 32.38 mg/kg), the plasma C_max of PCA was 119.95 ng/mL at 0.393 h, with a distribution half-life of 7.19 h and a markedly prolonged terminal half-life of 843.45 h. The AUC₀→∞ was 10,177.69 ng·h/mL [210]. Although differences in dose, formulation, and biological matrix preclude direct comparison with the pure compound, these results suggest that the presence of other plant constituents may influence the pharmacokinetics of PCA, possibly by modulating its absorption or metabolism. Regarding toxicity, PCA is generally considered safe at dietary levels, but high doses have been associated with mild hepatotoxic and nephrotoxic effects. In ICR mice, intraperitoneal administration of 500 mg/kg PCA significantly reduced glutathione (GSH) levels in liver and kidney, and increased plasma ALT and urea levels, suggesting oxidative stress-related toxicity. Moreover, subchronic exposure via drinking water (0.1% for 60 days) led to sustained renal GSH depletion and elevated ALT activity, without major systemic toxicity [211]. This change can be found on page 27 and on line 891.
6.5. Pharmacokinetic and toxicology Hydroxycinnamic acids are small phenolic acids abundant in plant foods. After oral intake, these compounds exhibit rapid but limited absorption, extensive first-pass metabolism, and efficient renal clearance. Absorption occurs via passive diffusion and likely via monocarboxylate transporters (e.g. MCT/SLC5A8) – sinapic acid in particular is reported to use a Na⁺‑dependent monocarboxylate transporter in the proximal gut [223].These acids are absorbed very quickly: in general, their absorption half‐lives are on the order of a few minutes (t_1/2≈0.07–0.08 h) and time-to-peak (T_max) is typically <1 h. Indeed most is taken up before reaching the colon. Gastric uptake has also been observed, consistent with short T_max. They are relatively hydrophilic and do not concentrate heavily in fat or cross membranes extensively. Some tissue uptake may occur in rodents: quantitative studies reveal that caffeic acid distributes to kidney, liver, muscle, lung, heart, spleen and testis. Ferulic acid showed even wider tissue presence in kidney, with smaller fractions in liver, lung, heart, spleen and brain [224]. On the other hand, quantitative tissue distribution data in humans are scarce. Acute and chronic toxicity studies indicate that hydroxycinnamic acids have low inherent toxicity at dietary or even nutraceutical doses. In human trials, very high supplementation has been well tolerated. For instance, 1000 mg/day of ferulic acid for six weeks led to substantial pharmacodynamic effects (improved lipid profiles) and caused no reported adverse effects and no changes in liver or kidney function tests [227]. Although this indicates a favorable safety profile and systemic exposure, formal toxicological limits in humans have yet to be established. This change can be found on page 29 and on line 1022. |
||
|
Comments 3: Some perspectives should be given. Response 3: We appreciate the reviewer’s suggestion to include future perspectives in the manuscript. In response, we have added a new paragraph at the end of the conclusion section to highlight potential research directions and translational opportunities arising from our review. This addition provides a broader context for the therapeutic potential of hydroxybenzoic and hydroxycinnamic acid derivatives and outlines key areas for future investigation. The new paragraph reads as follows: "From a translational perspective, these findings open several promising avenues for future research. One critical step would be the optimization of formulation and delivery strategies to improve the bioavailability and tissue targeting of these compounds. Additionally, more comprehensive preclinical and clinical studies are needed to validate their efficacy and safety in humans across various disease models. Given their multitargeted mechanisms of action, the development of combinatorial therapies that incorporate hydroxybenzoic or hydroxycinnamic acid derivatives with conventional treatments may also enhance therapeutic outcomes. Finally, their natural origin and favorable safety profile make them attractive candidates for inclusion in preventive strategies and nutraceutical formulations aimed at reducing the risk or delaying the onset of chronic diseases associated with inflammation, oxidative stress, and mitochondrial dysfunction." This change can be found on page 31 and on line 1165.
Comments 4: L 178: the sentence “4-HB is an active component in various plant extracts, such as Ageratina ringens, WHERE it contributes to both antioxidant effects and cardiovascular health benefits.” should be corrected. Response 4: The sentence has been corrected as follows: “4-HB is an active component in various plant extracts, such as Ageratina ringens. 4-HB has demonstrated significant antioxidant effects and cardiovascular health benefits.” This change can be found on page 13 and on line 195.
Comments 5: L 307: microorganism names should be in italic as before. In vitro, in vivo, ex vivo should be in italic all along the text. Response 5: It has been corrected. These changes can be found on page 16 and on line 353-355.
Comments 6: The chemical structures of all the substances that are discussed in this review should be presented in a separate figure Response 6: The chemical structures of all the compounds mentioned are presented in Table 1, which also provides an overview of their mechanisms of action and therapeutic effects. We chose to include the chemical structures in the table to facilitate understanding through a summary of both the chemical and biological characteristics of the compounds
|
||
Reviewer 2 Report
The authors provide a comprehensive review of benzoic and cinnamic acid derivatives, highlighting the biological activities of several of them and specifically addressing the mechanisms of action and therapeutic applications
The conclusions and perspectives are promising.
- ABSTRACT
Lines 12-15: Hydroxybenzoic and hydroxycinnamic acid............ from hydroxybenzoic and hydroxycinnamic acid, .........respectively(?), a type of benzoic acid in which one of...........
Changes to: Hydroxybenzoic and hydroxycinnamic acids............ from benzoic and cinnamic acids, ......... respectively, two type of acids in which one of...........
- INTRODUCTION
Lines 32-35: ........ Groups which structurally derive from the non-phenolic molecules of benzoic and cinnamic acid, respectively.
Obs.: The sentence makes no sense!
Line 38: ........ is separated by double bonded carbon pair [1,3].
Suggestion: ........ is separated by a carbon-carbon double bond.
Line 410-41: Derivatives of these compounds......... modifications of the phenol ring, usually by the addition of hydroxyl or alkoxyl (-O-R) substituents [1,5,8,11].
Change to: ......... modifications in the benzene ring, usually........
Line 86: Table 1. Mechanisms of action and therapeutic effects of ..........
Obs.: The mechanisms of action are shown in Figures 1 and 2. Table 1 shows the effects of.....
Suggestion: Table 1. Effects of ...........
Author Response
|
1. Summary |
|
|
|
We would like to thank the reviewer for the interest in this review. We also appreciate the useful recommendations to improve the manuscript.
|
||
|
2. Point-by-point response to Comments and Suggestions for Authors |
||
|
Comments 1: ABSTRACT Lines 12-15: Hydroxybenzoic and hydroxycinnamic acid............ from hydroxybenzoic and hydroxycinnamic acid, .........respectively(?), a type of benzoic acid in which one of........... Changes to: Hydroxybenzoic and hydroxycinnamic acids............ from benzoic and cinnamic acids, ......... respectively, two type of acids in which one of........... |
||
|
Response 1: Agree. It has been corrected. This change can be found on page 1 and on line 13.
|
||
|
Comments 2: INTRODUCTION Lines 32-35: ........ Groups which structurally derive from the non-phenolic molecules of benzoic and cinnamic acid, respectively. Obs.: The sentence makes no sense! |
||
|
Response 2: The sentence has been corrected as “From a structural perspective, phenolic acids and their derivatives fall under two main groups: hydroxybenzoic acids and hydroxycinnamic acids (Table 1) [5,10]. These groups are structurally derived from the non-phenolic molecules of benzoic and cinnamic acid, respectively.” This change can be found on page 1 and on line 33. |
||
|
Comments 3: Line 38: ........ is separated by double bonded carbon pair [1,3]. Suggestion: ........ is separated by a carbon-carbon double bond. Response 3: Agree. It has been corrected. This change can be found on page 1 and on line 39.
Comments 4: Line 410-41: Derivatives of these compounds......... modifications of the phenol ring, usually by the addition of hydroxyl or alkoxyl (-O-R) substituents [1,5,8,11]. Change to: ......... modifications in the benzene ring, usually........ Response 4: Agree. It has been corrected. This change can be found on page 1 and on line 40-41.
Comments 5: Line 86: Table 1. Mechanisms of action and therapeutic effects of .......... Obs.: The mechanisms of action are shown in Figures 1 and 2. Table 1 shows the effects of..... Suggestion: Table 1. Effects of ........... Response 5: It has been corrected. These changes can be found on page 3 and on line 98.
|
||
Reviewer 3 Report
The review titled Hydroxybenzoic and hydroxycinnamic acids derivatives: mechanisms of action and therapeutic applications by López-Herrador and co-workers deals with therapeutically application of hydrohycinnamic and hydrohybenzoic acid derivatives. Although the review has certain potential to be published in Antioxidants, the author must provide answers on following issues before acceptance:
- The title of the review is nor fully reflecting its content, since the word derivatives is a broad term that also includes synthetic derivatives of these molecules.
- The criteria for the choice of the compounds presented in manuscript is not clear, since there are other examples of these acids that also found significant therapeutically application. Therefore, the scope of the article must be extended to include more compounds.
- The values for observed biological activities should be added in table (such as IC50 for antitumor activities, etc), of course, where is that possible.
- The quality of the Figures 1 and 2 must be improved.
- The text must be checked carefully and corrected, for example, p in p-Coumaric acid should be in italic style, (3,4-Dihy-droxycinnamic acid in table should be presented above the structure, etc.
The review titled Hydroxybenzoic and hydroxycinnamic acids derivatives: mechanisms of action and therapeutic applications by López-Herrador and co-workers deals with therapeutically application of hydrohycinnamic and hydrohybenzoic acid derivatives. Although the review has certain potential to be published in Antioxidants, the author must provide answers on following issues before acceptance:
- The title of the review is nor fully reflecting its content, since the word derivatives is a broad term that also includes synthetic derivatives of these molecules.
- The criteria for the choice of the compounds presented in manuscript is not clear, since there are other examples of these acids that also found significant therapeutically application. Therefore, the scope of the article must be extended to include more compounds.
- The values for observed biological activities should be added in table (such as IC50 for antitumor activities, etc), of course, where is that possible.
- The quality of the Figures 1 and 2 must be improved.
- The text must be checked carefully and corrected, for example, p in p-Coumaric acid should be in italic style, (3,4-Dihy-droxycinnamic acid in table should be presented above the structure, etc.
Author Response
|
1. Summary |
|
|
|
We would like to thank the reviewer for the interest in this review. We also appreciate the useful recommendations to improve the manuscript.
|
||
|
2. Point-by-point response to Comments and Suggestions for Authors |
||
|
Comments 1: The title of the review is nor fully reflecting its content, since the word derivatives is a broad term that also includes synthetic derivatives of these molecules. |
||
|
Response 1: We appreciate the reviewer’s insightful comment regarding the clarity of the title. We agree that the term “derivatives” may be interpreted broadly to include synthetic or structurally modified analogs, whereas our review focuses primarily on naturally occurring derivatives of hydroxybenzoic and hydroxycinnamic acids with therapeutic relevance. In response, we have revised the title to more accurately reflect the scope and content of the manuscript. The updated title now reads: “Natural hydroxybenzoic and hydroxycinnamic acids derivatives: mechanisms of action and therapeutic applications” This change can be found on page 1 and on line 2.
|
||
|
Comments 2: The criteria for the choice of the compounds presented in manuscript is not clear, since there are other examples of these acids that also found significant therapeutically application. Therefore, the scope of the article must be extended to include more compounds. |
||
|
Response 2: We thank the reviewer for this constructive observation. We understand the importance of clarifying the selection criteria for the compounds included in the review. Our intention was not to provide an exhaustive catalog of all hydroxybenzoic and hydroxycinnamic acid derivatives, but rather to focus on a representative subset of well-characterized, naturally occurring compounds that have shown both mechanistic insight and therapeutic promise, particularly in relation to anti-inflammatory, antioxidant, and CoQ biosynthesis-related effects. |
||
|
Comments 3: Line 38: The values for observed biological activities should be added in table (such as IC50 for antitumor activities, etc), of course, where is that possible. Response 3: We thank the reviewer for this valuable suggestion. We fully agree that including quantitative data, such as IC₅₀ values for antitumor or other biological activities, would enrich the scientific content of the manuscript. However, there are limited or inconsistent reports of standardized IC₅₀ or EC₅₀ values, especially in the context of specific mechanisms discussed in our review (e.g., CoQ biosynthesis, mitochondrial function, or immune modulation). Due to the heterogeneity in experimental models, cell types, and methodologies, it was not possible to extract and compare such values in a meaningful and reliable way across compounds. Nevertheless, in order to enhance the translational value of the manuscript, we have introduced pharmacokinetic information for each compound. These data are now included in a new section within each compound, titled “Pharmacokinetic and toxicology”.
2.4. Pharmacokinetic and toxicology In vivo studies in rabbits showed that oral doses ranging from 100 to 1500 mg/kg of 4HB resulted in a total urinary recovery of 84% to 104%, indicating efficient absorption and rapid renal elimination. Metabolites conjugated with glucuronic acid (10–35%) and sulfate (4–7%) were identified in the urine, and all metabolite concentrations returned to baseline within 24 hours. These findings suggest that 4-HBA has good oral bioavailability and is efficiently excreted via the urinary route [164]. 4-Hydroxybenzoic acid also exhibits low acute toxicity in mammals, with oral and dermal LD50 values exceeding 2000 mg/kg in rats and rabbits, respectively, and an inhalation LC50 in rats greater than 0.47 mg/L (dust). In human research, 4-HBA has been administered orally without adverse effects: participants received up to 5 grams every 6 hours over 24 hours, or a total of 26 grams over 28 hours, diluted in drinking water [165]. Regarding environmental toxicity, 4HB shows moderate to low toxicity toward aquatic organisms. The 48-hour EC50 for Daphnia magna ranges from 173 to 1690 mg/L (median 932 mg/L), while the 96-hour EC50 for algae is reported at 42.8 mg/L. Overall, these data suggest that 4HB poses limited toxic risk to terrestrial and aquatic organisms, although precaution is warranted in ecologically sensitive aquatic environments [166]. This change can be found on page 15 and on line 293.
3.5. Pharmacokinetic and toxicology Not much information is available on the pharmacokinetic properties of BRA. In fact, a single article conducted in our laboratory determined that the plasma half-life of BRA after intravenous administration (50 mg/kg) is about 200 minutes, with the maximum concentration being reached five minutes after administration (1.63x105 ng/ml). These results leave an area under the curve of 3.5x1'6 ng/ml/min, with a volume of distribution of 1.42 × 105 (mg/kg)/(ng/ml) and a clearance (Cl) of 5.47 × 105 (mg/kg)/(ng/ml). Accordingly, sustained administration of BRA would be necessary to achieve a therapeutically relevant plasma concentration [35]. Regarding toxicity, available data suggest that BRA is relatively well tolerated at moderate to high doses in animal models. In rats, oral administration of up to 6000 mg/day for 16 days in humans resulted primarily in glucuronide and sulfate conjugates in urine, indicating efficient phase II metabolism [180]. In reproductive toxicity studies, subcutaneous administration of up to 380 mg/kg to pregnant rats did not affect implantation rates, fetal weight, resorptions, or malformations [181]. However, a higher cumulative dose (428 mg/kg followed by 214 mg/kg) was associated with a reduction in serum calcium and a 33% rate of fetal toxicity, including minor skeletal malformations. These findings were not statistically analyzed, and the authors noted that the effects differed from those typically observed with salicylate exposure [182]. Overall, BRA appears to have a low acute toxicity profile, although potential developmental effects at high doses warrant further investigation. This change can be found on page 18 and on line 454.
4.4. Pharmacokinetic and toxicology A study using ultra-high-performance liquid chromatography tandem mass spectrometry (LC-MS/MS) demonstrated that after oral administration of vanillic acid at doses of 2, 5, and 10 mg/kg in rats, the plasma concentration peaks were observed at 0.42 ± 0.09, 0.73 ± 0.21, and 0.92 ± 0.28 μg/ml, respectively, within 0.55-0.64 hours. The oral bioavailability was calculated to be between 25.3% and 36.2% [191]. The encapsulation of vanillic acid with hydroxypropyl-beta-cyclodextrin (HPBCD) significantly improved its solubility and bioavailability. The Tmax and AUC for the HPBCD-vanillic acid complex were 2 hours and 253.46 ± 3.42, respectively, suggesting enhanced pharmacokinetic properties compared to the native form [192]. To date, no significant toxic effects have been reported for VA, except in some microbial models, where its antimicrobial properties can inhibit the growth of certain microorganisms. As mentioned in the text, VA and its isomers exhibit antimicrobial activity against a variety of pathogens. In terms of systemic toxicity, studies in Wistar rats have shown that oral administration of VA at a dose of 1000 mg/kg/day for two weeks does not produce significant alterations in hematological or biochemical parameters, suggesting a relatively safe toxicological profile at this dosage [193]. This change can be found on page 23 and on line 687.
5.4. Pharmacokinetic and toxicology A study using a validated and sensitive LC-MS/MS method demonstrated that after oral administration of protocatechuic acid at a dose of 50 mg/kg in mice, the plasma concentration peaked at 73.6 μM within 4.87 minutes. The compound showed rapid absorption with an initial half-life of 2.9 minutes and a terminal half-life of 16 minutes. PCA remained detectable in plasma for up to 8 hours, and the area under the plasma concentration–time curve (AUC₀→₈h) was 1456 μM·min, indicating fast systemic exposure and two-phase elimination kinetics [208]. In the same study, PCA was also detected in human plasma at low ng/mL concentrations after oral intake of 60 g/day of black raspberry (BRB) powder for 21 days, reflecting its formation as a microbial metabolite of dietary anthocyanins [208]. This observation is further supported by other studies showing the presence of PCA and related metabolites in urine following oral administration of 500 mg of uniformly ¹³C-labeled cyanidin-3-glucoside to a cohort of human subjects [209]. Additional data from a separate pharmacokinetic study using Hedyotis diffusa Willd extract—a traditional medicinal plant naturally containing PCA among other phenolics—showed a distinct kinetic profile. Following oral administration of 4.837 g/kg of the extract in rats (providing a calculated PCA dose of 32.38 mg/kg), the plasma C_max of PCA was 119.95 ng/mL at 0.393 h, with a distribution half-life of 7.19 h and a markedly prolonged terminal half-life of 843.45 h. The AUC₀→∞ was 10,177.69 ng·h/mL [210]. Although differences in dose, formulation, and biological matrix preclude direct comparison with the pure compound, these results suggest that the presence of other plant constituents may influence the pharmacokinetics of PCA, possibly by modulating its absorption or metabolism. Regarding toxicity, PCA is generally considered safe at dietary levels, but high doses have been associated with mild hepatotoxic and nephrotoxic effects. In ICR mice, intraperitoneal administration of 500 mg/kg PCA significantly reduced glutathione (GSH) levels in liver and kidney, and increased plasma ALT and urea levels, suggesting oxidative stress-related toxicity. Moreover, subchronic exposure via drinking water (0.1% for 60 days) led to sustained renal GSH depletion and elevated ALT activity, without major systemic toxicity [211]. This change can be found on page 27 and on line 891.
6.5. Pharmacokinetic and toxicology Hydroxycinnamic acids are small phenolic acids abundant in plant foods. After oral intake, these compounds exhibit rapid but limited absorption, extensive first-pass metabolism, and efficient renal clearance. Absorption occurs via passive diffusion and likely via monocarboxylate transporters (e.g. MCT/SLC5A8) – sinapic acid in particular is reported to use a Na⁺‑dependent monocarboxylate transporter in the proximal gut [223].These acids are absorbed very quickly: in general, their absorption half‐lives are on the order of a few minutes (t_1/2≈0.07–0.08 h) and time-to-peak (T_max) is typically <1 h. Indeed most is taken up before reaching the colon. Gastric uptake has also been observed, consistent with short T_max. They are relatively hydrophilic and do not concentrate heavily in fat or cross membranes extensively. Some tissue uptake may occur in rodents: quantitative studies reveal that caffeic acid distributes to kidney, liver, muscle, lung, heart, spleen and testis. Ferulic acid showed even wider tissue presence in kidney, with smaller fractions in liver, lung, heart, spleen and brain [224]. On the other hand, quantitative tissue distribution data in humans are scarce. Acute and chronic toxicity studies indicate that hydroxycinnamic acids have low inherent toxicity at dietary or even nutraceutical doses. In human trials, very high supplementation has been well tolerated. For instance, 1000 mg/day of ferulic acid for six weeks led to substantial pharmacodynamic effects (improved lipid profiles) and caused no reported adverse effects and no changes in liver or kidney function tests [227]. Although this indicates a favorable safety profile and systemic exposure, formal toxicological limits in humans have yet to be established. This change can be found on page 29 and on line 1022.
Comments 4: The quality of the Figures 1 and 2 must be improved. Response 4: We have revised and replaced Figures 1 and 2 with high-resolution versions to ensure optimal clarity and legibility.
Comments 5: The text must be checked carefully and corrected, for example, p in p-Coumaric acid should be in italic style, (3,4-Dihy-droxycinnamic acid in table should be presented above the structure, etc. Response 5: We have thoroughly reviewed the entire manuscript to correct formatting issues. |
||
Round 2
Reviewer 1 Report
The paper has been significantly improved and I find answers to my comments and questions.
Not any more comment.
Reviewer 3 Report
I believe that, after all changes, this manuscript could be suitable for the publication.
I believe that, after all changes, this manuscript could be suitable for the publication.